# Tbx5a lineage tracing shows cardiomyocyte plasticity during zebrafish heart regeneration

Héctor Sánchez-Iranzo[1], María Galardi-Castilla[1], Carolina Minguillón[2,7], Andrés Sanz-Morejón[1,3], Juan Manuel González-Rosa[1,4], Anastasia Felker[5], Alexander Ernst[3], Gabriela Guzmán-Martínez[6], Christian Mosimann [5] & Nadia Mercader [1,3]

During development, mesodermal progenitors from the first heart field (FHF) form a primitive cardiac tube, to which progenitors from the second heart field (SHF) are added. The contribution of FHF and SHF progenitors to the adult zebrafish heart has not been studied to date. Here we find, using genetic *tbx5a* lineage tracing tools, that the ventricular myocardium in the adult zebrafish is mainly derived from *tbx5a*[+] cells, with a small contribution from *tbx5a*[−] SHF progenitors. Notably, ablation of ventricular *tbx5a*[+]-derived cardiomyocytes in the embryo is compensated by expansion of SHF-derived cells. In the adult, *tbx5a* expression is restricted to the trabeculae and excluded from the outer cortical layer. *tbx5a*-lineage tracing revealed that trabecular cardiomyocytes can switch their fate and differentiate into cortical myocardium during adult heart regeneration. We conclude that a high degree of cardiomyocyte cell fate plasticity contributes to efficient regeneration.

---

[1] Development of the Epicardium and Its Role during Regeneration Group, Centro Nacional de Investigaciones Cardiovasculares (CNIC-ISCIII), Melchor Fernández Almagro 3, 28029 Madrid, Spain. [2] CSIC-Institut de Biologia Molecular de Barcelona Parc Científic de Barcelona C/ Baldiri i Reixac, 10 08028 Barcelona, Spain. [3] Institute of Anatomy, University of Bern, 3000 Bern 9, Switzerland. [4] Cardiovascular Research Center, Massachusetts General Hospital and Harvard Medical School, Boston, MA 02114, USA. [5] Institute of Molecular Life Sciences, University of Zürich, 8057 Zürich, Switzerland. [6] Hospital Universitario La Paz, IdiPAZ, 28046 Madrid, Spain. [7] Present address: Barcelonabeta Brain Research Center, Pasqual Maragall Foundation, 08005 Barcelona, Spain. Correspondence and requests for materials should be addressed to N.M. (email: nadia.mercader@ana.unibe.ch)

The zebrafish is an established model organism to study heart regeneration, in which pre-existing cardiomyocytes proliferate to replace the lost myocardium[1–3]. The zebrafish myocardium is formed by an inner trabecular layer, a thin primordial layer, and an outer cortical layer. During embryonic development, cells from the primordial layer give rise to trabeculae[4]. Later, in the juvenile, trabecular cardiomyocytes breach the primordial layer and form the cortical myocardium[5,6]. Previous clonal analysis suggested that during adult heart regeneration, the cortical myocardium is rebuilt by cardiomyocytes from the same layer[5], suggesting some degree of commitment to a particular myocardial layer. A higher degree of plasticity had been observed in the embryonic heart, whereby atrial cardiomyocytes were reported to be able to regenerate the cardiac ventricle[7].

The vertebrate heart is formed from mesodermal precursor cells derived from the first and second heart fields, also called FHF and SHF, respectively. Cells from the FHF form the embryonic heart tube to which cells from the SHF are added to allow further growth. FHF and SHF progenitors have also been described in the zebrafish[8–12]. Regulatory elements from the *draculin* (*drl*) gene have been used to label a population of lateral plate mesoderm progenitors that give rise to cardiomyocytes of the primitive heart tube, providing a genetic marker to trace FHF-derived myocardium[11]. *Latent tgf beta binding protein 3* (*ltbp3*) expression marks a distal *drl*-negative domain and has been proposed as a SHF marker[12]. Of note, neither *drl* nor *ltbp3* are established FHF and SHF markers in other vertebrates including mammals[11].

In the mouse, retrospective clonal analysis revealed that FHF-derived cells predominantly give rise to the left ventricle, whereas the SHF yields the right ventricle and large parts of the atria[13,14]. Further studies on the contribution of cardiac precursor cells to the mammalian heart come from genetic lineage tracing approaches, which are based on following the fate of cells that expressed a particular gene at a given time point of embryonic development[15–17]. A key gene used to trace the fate of cardiac precursor cells is *Tbx5*, encoding a T-box transcription factor, which serves as an early cardiac marker expressed in the developing heart tube[18,19]. Genetic lineage tracing analysis of *Tbx5*-expressing cells has revealed a strong contribution to the left ventricular myocardium, recapitulating the fate of FHF progenitors in this cardiac chamber[20].

Zebrafish have two *Tbx5*-encoding genes, *tbx5a* and *tbx5b*; *tbx5a* expression levels during heart development are higher than those of *tbx5b*, and *tbx5a* mutants recapitulate key phenotypes of mammalian *Tbx5* perturbations[21–23]. We thus decided to generate genetic tools to study the contribution of the FHF to the adult ventricle during homeostasis and regeneration in the zebrafish using regulatory sequences of *tbx5a*. We describe that in zebrafish the early heart tube is formed by *tbx5a*-derived cardiomyocytes to which SHF-derived *tbx5a*⁻ cardiomyocytes are added. Even though their contribution to the adult heart is minimal, SHF-derived cells can give rise to a fully functional ventricle if *tbx5a*-derived cells are genetically ablated. In the adult heart, *tbx5a* expression is persistent only in the trabecular myocardium, allowing to distinguish this layer from the cortical myocardium. Genetic fate mapping of trabecular *tbx5a*⁺ cells shows that these cells contribute to regenerate the outer cortical layer. Together, our findings show a high degree of cardiomyocyte fate plasticity during zebrafish heart regeneration, during both embryonic stages and in the adult.

## Results

### Contribution of *tbx5a*⁺ cardiomyocytes to the adult ventricle.
We used the fluorescent bacterial artificial chromosome (BAC)

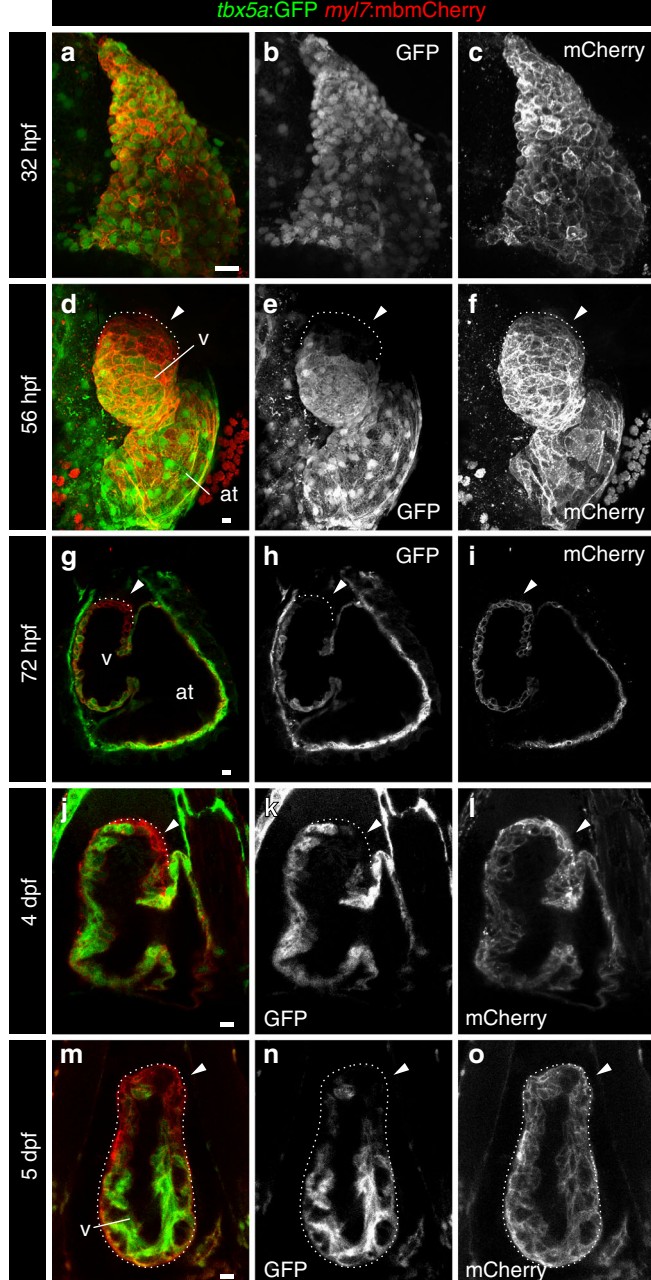

**Fig. 1** Expression profile of *tbx5a*-positive cardiomyocytes in embryonic zebrafish hearts. **a–f** Whole-mount immunofluorescence of *tbx5a:GFP;myl7: mbmCherry* double transgenic zebrafish hearts at 32 **a–c** (n = 5/5) and 56 hours postfertilisation (hpf) **d–f** (n = 3/3). **g–l** Confocal optical sections of *tbx5a:GFP;myl7:mbmCherry* hearts at 72 hpf **g–i** (n = 5/5), 4 days postfertilisation (dpf) **j–l** (n = 9/9), and 5 dpf (**m–o**; n = 6/6). GFP (green) labels *tbx5a*⁺ cells and mCherry (red) marks cells expressing the pan-myocardial marker *myosin light chain 7* (*myl7*). Shown are ventral views, cranial is to the top. At 32 hpf all cardiomyocytes are *tbx5a:GFP*⁺ but at 56, 72 hpf, 4, and 5 dpf *tbx5a:GFP*⁻ cardiomyocytes can be observed in the distal ventricle (arrowheads). The atrioventricular canal and large portions of the atrium are also GFP⁺. at, atrium; v, ventricle; Scale bars, 10 μm

transgenic reporter line *tbx5a:GFP*, driving green fluorescent protein (GFP) expression under the control of the *tbx5a cis*-regulatory elements[24]. The overall GFP expression pattern of *tbx5a:GFP* recapitulated that of endogenous *tbx5a* expression in embryos (Supplementary Fig. 1a, b; n = 7/7). We analysed the

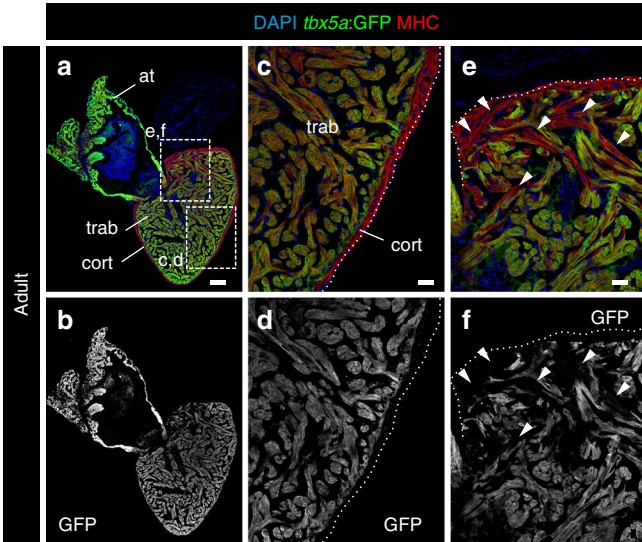

**Fig. 2** Expression profile of *tbx5a*-positive cardiomyocytes in adult zebrafish hearts. **a**, **b** Sagittal sections through *tbx5a:GFP* adult uninjured heart immunostained with GFP (green) and Myosin Heavy Chain (MHC; red). Nuclei are counterstained with DAPI (blue). **c**–**e** Zoomed views of boxed area in **a**. **b**, **d**, **f** Single channels for GFP. The trabecular myocardium is *tbx5a:GFP*+, whereas the cortical layer is *tbx5a:GFP*−. Note *tbx5a:GFP*− cardiomyocytes (arrowheads) in the basal part of the ventricle close to the atrioventricular canal ($n = 13/13$). at, atrium; cort, cortical layer; trab, trabecular layer. Scale bars, **a**, **b** 100 μm and **c**–**f** 25 μm

expression of *tbx5a:GFP* in the early heart tube, before the addition of SHF progenitors. At this stage, the heart tube is formed by a single sheet of cardiomyocytes named the primordial layer[5]. Homogeneous *tbx5a:GFP* expression was detected in the whole embryonic myocardium at 32 hours postfertilisation (hpf) (Fig. 1a–c; $n = 5/5$). After this stage, at 56 hpf, (Fig. 1d–f; $n = 3/3$), a GFP− region of myocardium started to become visible in the cranial portion of the heart tube, which remained reporter-negative at 72 hpf (Fig. 1g–i; Supplementary Movie 1, $n = 5/5$), 4 days postfertilisation (dpf) (Fig. 1j–l; $n = 9/9$), and at 5 dpf (Fig. 1m–o; $n = 6/6$), concomitant with the reported addition of SHF progenitors to the ventricular myocardium during these stages[8,12]. The GFP expression pattern was confirmed by *gfp* messenger RNA in situ hybridisation on heart sections of embryos at 2–5 dpf (Supplementary Fig. 1c–g). Comparison of the expression domains at 56 and 72 hpf between *tbx5a:GFP* and the previously reported FHF marker *drl:mCherry* revealed an overlap of both domains in the proximal part of the ventricle close to the atrioventricular canal (AVC) and in the outer curvature of the ventricle (Supplementary Fig. 2a–h and Supplementary Movies 2 and 3; 56 hpf, $n = 5/5$; 72 hpf, $n = 7/7$). Of note, the expression domain of *tbx5a:GFP* was more extensive toward the distal myocardium. This might indicate that *drl* expression is downregulated faster in FHF-derived cells than *tbx5a*, or that part of the proximal SHF-derived myocardium expresses *tbx5a*. Nonetheless, a clear *tbx5a:GFP/drl:mCherry* double-negative domain was present in the inner curvature and distal-most portion of the larval ventricle (Supplementary Fig. 2a–h and Supplementary Movies 2 and 3). This *tbx5a:GFP*− region was further expressing transiently injected *ltbp3:TagRFP-2A-Cre*[12], indicating SHF-derived ventricular myocardium (Supplementary Fig. 2i–l and Supplementary Movie 4). Thus, in the embryonic heart, *tbx5a* expression marks cardiomyocytes in the primitive heart tube and is absent from a population of distally located ventricular cardiomyocytes, which from their expression pattern and

temporal appearance match with previous descriptions of the most distal SHF-derived cells[12].

We next sought to chart *tbx5a* expression in the adult zebrafish heart. During zebrafish heart maturation, a third myocardial layer forms enveloping the primordial layer, named cortical layer. Clonal analysis suggested that these myocardial layer derives from trabecular cardiomyocytes[5]. We still detected high levels of endogenous *tbx5a* mRNA and *tbx5a:GFP* expression in the adult zebrafish ventricle (Fig. 2a, b and Supplementary Fig. 3; $n = 8/8$). *tbx5a:GFP* expression was restricted to the trabecular myocardium and absent from the cortical layer (Fig. 2c, d), coinciding with the recently reported pattern obtained using an enhancer fragment located 16 kb upstream of the *tbx5a* transcription start site[25]. Expression spread through most of the trabecular myocardium, with a gradient from the apex (high expression) to the basal part close to the bulbus arteriosus (BA) (lower expression levels). In addition, we observed a region in the basal ventricle, close to the atrium, in which we detected GFP− trabecular cardiomyocytes. No GFP expression can be detected in these cells beyond background and we therefore refer to this myocardial territory as basal *tbx5a*− region (Fig. 2e, f; $n = 13/13$; Supplementary Fig. 4).

*tbx5a*− cardiomyocytes in the adult heart could derive from embryonic *tbx5a*+ cells that switched off their expression or from progenitor cells that never expressed *tbx5a*. To test these possibilities, we generated transgenic lines for inducible genetic fate mapping of *tbx5a*-expressing cells (Fig. 3a). In double-transgenic *tbx5a:mCherry-p2A-CreER*$^{T2}$;*ubb:loxP-lacZ-STOP-loxP-GFP*[26] embryos, mCherry labels *tbx5a*-expressing cells and GFP labels the progeny of *tbx5a*+ cells that expressed CreER$^{T2}$ recombinase at the time of 4-Hydroxytamoxifen (4-OHT)-induced recombination. This transgenic combination was not leaky and GFP-expressing cells were visible only upon 4-OHT administration (Fig. 3b; $n = 5/5$). When recombined during embryogenesis, GFP+ cells were found in the atrium and ventricle of the larval heart but never in the distal portion of the ventricular myocardium (Fig. 3c–f; 8/8; Supplementary Fig. 5 and Supplementary Movie 5, $n = 9/9$). Few epicardial cells were also *tbx5a*-derived, consistent with *tbx5a:tdTomato* expression observed in a subset of embryonic epicardial cells (Supplementary Fig. 6; $n = 6/7$). When analysed in adult hearts, the cortical layer was mCherry−, in agreement with our observations using the reporter *tbx5a:GFP*. Nonetheless, most of the cortical layer was GFP+ and thus derived from a *tbx5a*+ embryonic cell population that switched off *tbx5a* expression (Fig. 3g–n; $n = 5/5$; Supplementary Fig. 7, 8).

Akin to the GFP pattern observed in *tbx5a:GFP* adult hearts, a basal domain of the ventricle close to the BA was also mCherry− (Fig. 3g–j and 3o–r, and Supplementary Fig. 7, 8; $n = 5/5$). Furthermore, this domain was also GFP−, suggesting that it derives from the *tbx5a*− distal embryonic ventricle. Thus, we propose that *tbx5a*+-derived cells in the adult heart predominantly comprise the FHF derivatives of the embryonic heart and the *tbx5a*− domain in the basal ventricle close to the atrium and BA is SHF-derived. Taken together, our genetic lineage tracing of embryonic myocardial populations into adulthood in zebrafish reveal a fate compartmentalisation of *tbx5a*− and *tbx5a*+-derived cardiomyocytes starting in embryogenesis that results in distinguishable ventricle domains in the absence of septation.

**SHF progenitors compensate ablated *tbx5a*+ cardiomyocytes.** Previous results have indicated that atrial cardiomyocytes acquire a ventricular phenotype upon injury in the embryonic heart[7]. To explore whether SHF progenitors can compensate for a loss of FHF-derived cells, we generated *vmhcl:loxP-tagBFP-loxP-mCherry-NTR*, a transgenic line to specifically mark and ablate

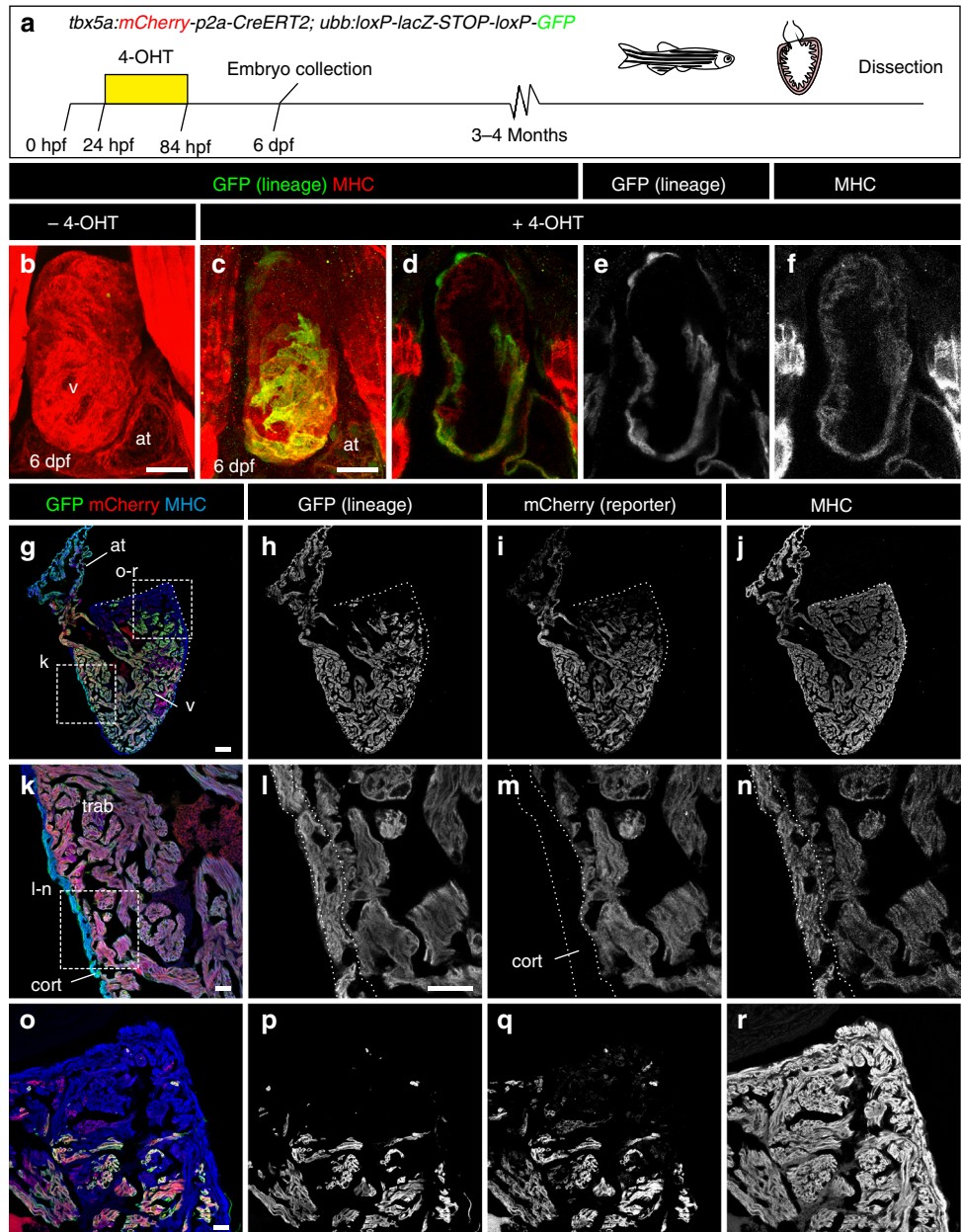

**Fig. 3** Fate mapping of *tbx5a*-derived cells during cardiac development. **a** *tbx5a:mCherry-p2A-CreER^T2;ubb:loxP-LacZ-STOP-loxP-GFP* hearts fixed at different developmental stages. mCherry marks *tbx5a*-expressing cells; GFP marks *tbx5a*-derived cells. **b–f** Whole-mount ventral view of hearts at 6 days postfertilisation (dpf) stained for GFP (green) and Myosin Heavy Chain (MHC, red). **b** In the absence of 4- Hydroxytamoxifen (4-OHT) administration, no GFP+ cells are visible (*n* = 5/5). **c–f** 4-OHT was added from 24 to 84 hours postfertilisation (hpf). GFP expression is observed in the proximal part of the ventricle; *n* = 8/8. In some cases, GFP expression was also found in epicardial cells located in the distal part of the ventricle. **g–r** Immunofluorescence staining of adult heart sections recombined as in **c**. Shown are merged and single channels for GFP (green), mCherry (red), and anti-MHC staining (blue); *n* = 5/5. at, atrium; cort, cortical layer; trab, trabecular layer; v, ventricle. Scale bars, **g** 100 μm and **b**, **c**, **k**, **l**, **o** 25 μm

ventricular cardiomyocytes using the *ventricular myosin heavy chain like* (*vmhcl*) *cis*-regulatory elements (Fig. 4a; Supplementary Fig. 9; *n* = 10/10). *vmhcl* is expressed exclusively in ventricular and not atrial cardiomyocytes[27]. We crossed *vmhcl:loxP-tagBFP-loxP-mCherry-NTR* with *tbx5a:CreER^T2*. This allowed to lineage-trace *tbx5a+* (mCherry+) and *tbx5a−* ventricular cardiomyocytes (tagBFP+). Furthermore, this genetic strategy allows the ablation of *tbx5a+* ventricular cardiomyocytes by adding Metronidazole (Mtz) to induce cell death in nitroreductase (NTR)-expressing cells[28]. Administration of 4-OHT from 24 to 48 hpf induced the expression of mCherry-NTR in *tbx5a*-derived cardiomyocytes in the trabecular and primordial layer of the embryonic ventricle,

with the exception of the distal-most part at the cranial pole (Fig. 4b, c, *n* = 7/7). In adult hearts, most ventricular cardiomyocytes were mCherry+. Only a small portion of cardiomyocytes close to the BA and AVC was tagBFP+ (Fig. 4d, e), coinciding with the results obtained doing lineage tracing using *tbx5a:mCherry-p2a-CreER^T2;ubb:loxP-lacZ-STOP-loxP-GFP*. To specifically ablate mCherry+ cardiomyocytes we administered Mtz from 4 to 7 dpf (Supplementary Fig. 10). Surprisingly, 30 days later, a complete ventricle had regenerated from tagBFP+ cardiomyocytes, suggesting that SHF-derived ventricular cardiomyocytes are able to compensate the genetic ablation of the FHF-derived ventricular myocardium (*n* = 6/6; Fig. 4f–h and

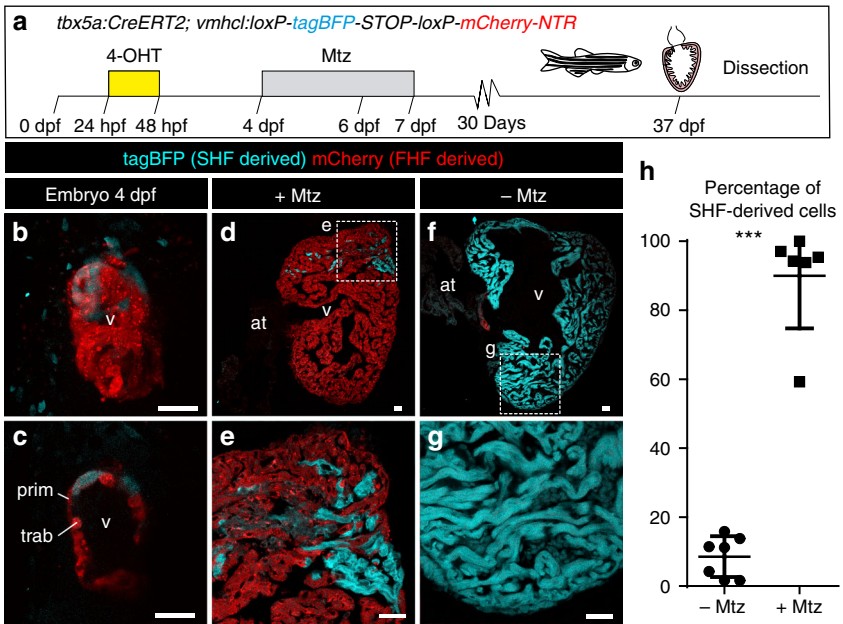

**Fig. 4** Genetic ablation of *tbx5a*-derived ventricular cardiomyocytes. **a** *tbx5a*+ ventricular cardiomyocytes were genetically ablated in *tbx5a:CreER^T2;vmhcl: loxP-tagBFP-loxP-mCherry-NTR* double transgenic zebrafish. Recombination was induced by administration of 4-Hydroxytamoxifen (4-OHT). Cell ablation was induced by administration of Metronidazole (Mtz) from 4 to 7 days postfertilisation. **b, c** Ventral views of larval hearts at 4 dpf (maximal projection and optical section, respectively). Anterior is to the top. The proximal ventricle, including primordial layer and trabeculae, is completely mCherry+ and the distal ventricle is blue (tagBFP+); n = 7/7. **d, e** Sagittal section of the ventricle of an adult recombined heart. Most cells are mCherry+. Only the *tbx5a*− region is tagBFP. n = 7/7. **f, g**, Sagittal section of a Mtz-treated fish. Most of the cardiomyocytes are tagBFP+; n = 6/6. **h**, Quantification of the percentage of myocardium that is tagBFP+ (SHF-derived), mean±SD; *** P < 0.0001 by two-tailed unpaired *t*-test. at, atrium; prim, primordial layer; SHF, second heart field; trab, trabeculae; v, ventricle. Scale bars, 25 μm

Supplementary Fig. 11). In the experimental setup all atrial cardiomyocytes are *tbx5a* lineage-derived and thus would become mCherry+ if they contributed to the ventricle (Supplementary Fig. 5 and Supplementary Movie 5, n = 9/9). Thus, our results reveal that ventricular *tbx5a*− SHF-derived cells compensated for the loss of *tbx5a*+ ventricular cardiomyocytes. To confirm ablation of *tbx5a*-derived mCherry-NTR expressing cells, we performed terminal deoxynucleotidyl transferase (TdT)-mediated dUTP nick end labeling (TUNEL) staining to detect apoptosis (Fig. 5a–g; n = 8 larvae + Mtz group and n = 8 −Mtz group) and detected a significant increase of TUNEL+ mCherry+ trabecular and primordial cardiomyocytes in the Mtz-treated group. In agreement, tagBFP+ ventricular cardiomyocytes incorporated significantly more 5-bBromo-2′-deoxyuridine (BrdU) upon Mtz treatment, indicating proliferation of this cell population (Fig. 5h–m).

Ablation of *tbx5a*-derived ventricular cardiomyocytes led to a transient impairment of cardiac function. At 7 dpf, right after the Mtz treatment, the ventricles revealed an irregular shape and displayed diminished pumping efficiency (Supplementary Fig. 12a–e and Supplementary Movie 6). However, mortality was not significantly increased (Supplementary Fig. 12f). Indeed, in the adults, the regenerated hearts revealed normal ventricular pumping efficiency (Supplementary Fig. 12g). Taken together, our results demonstrate that the loss of FHF-derived cardiomyocytes is compensated by an expansion of SHF-derived cardiomyocytes.

**Cardiomyocyte plasticity during adult heart regeneration**. We next explored whether this observed cardiomyocyte plasticity remains in the adult. Upon cryoinjury, cardiomyocytes at the site of injury proliferate and rebuild the lost cortical and trabecular myocardium[1,2]. Using our transgenic tools, which genetically discriminate between cortical and trabecular cardiomyocytes, we investigated the plasticity of trabecular cardiomyocyte cell fate during regeneration. First, we analysed the expression of *tbx5a*: GFP during heart regeneration using cryoinjury (Supplementary Fig. 13a). *tbx5a*:GFP was highly expressed in uninjured trabeculae at all stages analysed. At 1 days postinjury (dpi), no GFP expression could be found in the cortical layer (Supplementary Fig. 13b–f; n = 4/4). In contrast, at 3 dpi (Supplementary Fig. 13g–k; n = 4/5), 7 dpi (Supplementary Fig. 13l–p; n = 3/3), and 21 dpi (Supplementary Fig. 13q–u; n = 4/4), few *tbx5a*:GFP+ cardiomyocytes could be observed in the cortical layer close to the injured area. Nonetheless, at 130 dpi, the regenerated cortical myocardium was *tbx5a*:GFP− (Supplementary Fig. 13v–z; n = 4/ 4). This transient expression of *tbx5a* in cortical cardiomyocytes can have two explanations: (a) *tbx5a* is re-expressed transiently in the cortical layer during regeneration, or (b) trabecular cardiomyocytes are contributing to the new cortical layer.

To evaluate these two possibilities, *tbx5a*-derived cells were genetically traced using *tbx5a:mCherry-p2A-CreER^T2;ubb:loxP-STOP-loxP-GFP* double-transgenic fish (Fig. 6a). Upon recombination, GFP expression was restricted to the trabeculae in uninjured hearts (Fig. 6b–g; n = 7/7) and we only rarely detected GFP+ cells in the cortical myocardium (two cells in one out of six completely sectioned hearts, Supplementary Fig. 14a–f). By contrast, we observed GFP+/mCherry− cells in the regenerated cortical layer at 21 dpi (Fig. 6h–m; n = 6/7) and 90 dpi (Fig. 6n, o–s; n = 6/6), indicating that these cells derived from *tbx5a*+ cells had switched off *tbx5a* expression. In the remote area, GFP+ cells remained restricted to the trabecular layer even upon cryoinjury (Fig. 6t–x; n = 5/6). Based on the percentage of trabecular cardiomyocytes that are GFP+, the recombination efficiency was estimated to be 10 ± 6%. In the newly formed cortical layer, 7 ±

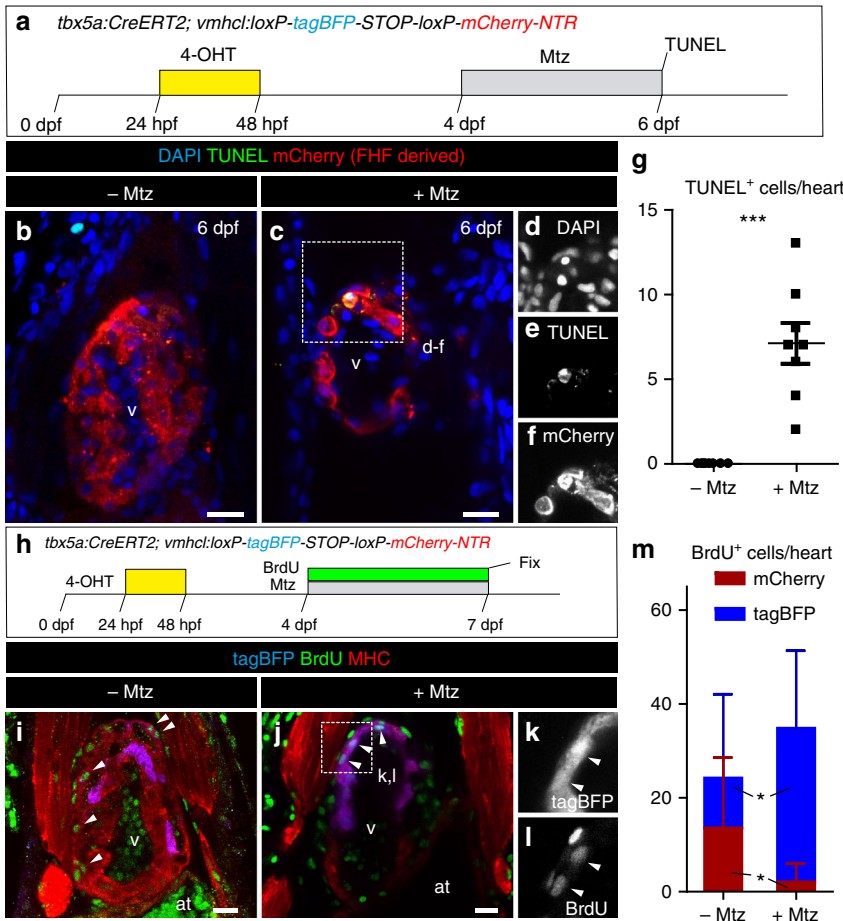

**Fig. 5** Apoptosis and cell proliferation upon genetic ablation of *tbx5a*-derived ventricular cardiomyocytes. **a** *tbx5a*+ ventricular cardiomyocytes were genetically ablated in *tbx5a:CreER^T2;vmhcl:loxP-tagBFP-loxP-mCherry-NTR* double transgenic zebrafish. Recombination was induced by administration of 4-Hydroxytamoxifen (4-OHT). Cell ablation was induced by administration of Metronidazole (Mtz) from 4 to 7 days postfertilisation (dpf). **b, c** Optical sections of 6 dpf fish that had been treated with 4-OHT and Mtz as indicated in **a** (**b**) or only with 4-OHT **c** immunostained for mCherry (red) and terminal deoxynucleotidyl transferase (TdT)-mediated dUTP nick end labeling (TUNEL) (green). Note that some rounded mCherry+ cells are TUNEL+. **d–f** Single channels of selected area in **c**. **g** Quantification of the number of TUNEL+ trabecular and cortical mCherry+ cardiomyocytes per heart from animals of the +Mtz ($n = 8$) and −Mtz ($n = 8$) group, mean ± SD; ***$P = 0.0004$ by Mann–Whitney non-parametric *t*-test. **h** Schematic representation of the 5-Bromo-2′-deoxyuridine (BrdU) treatment to assess proliferation. **i, j** Fish were treated with 4-OHT and BrdU **i** or with 4-OHT, Mtz, and BrdU **j**. **k, l** Single channels of the boxed area in **j**. **m** Quantification of BrdU+/mCherry+ and BrdU+/tagBFP+ cells per heart. Shown are means ± SD ($n = 8$ for −Mtz hearts and $n = 11$ for +Mtz hearts, from two separate independent experiments) *$P = 0.0240$ for tagBFP+ cells and $P = 0.0371$ for mCherry+ cells by two-tailed *t*-test. at, atrium; hpf, hours postfertilization; v, ventricle. Scale bars, 25 μm

4% of cardiomyocytes are GFP+ ($n = 10$). Therefore, trabeculae might contribute to a higher extent to the regenerating cortical layer than estimated by GFP signal detection.

To confirm that these cells came from *tbx5a*+ trabeculae and were not produced by leaky recombination in response to injury, we repeated the experiment in the absence of 4-OHT administration; we found no GFP+-recombined cells within the regenerated area (Supplementary Fig. 14g–q; $n = 3/3$). We further excluded that the result is caused by 4-OHT remaining in the body at the moment of injury through treating zebrafish with 4-OHT 10 and 11 days before the injury (Supplementary Fig. 14r–w; $n = 5/5$). Taken together, our results reveal that trabeculae contribute to the regeneration of cortical myocardium. This process occurs concomitantly to the downregulation of *tbx5a* expression, suggesting that trabecular cardiomyocytes change their specification and become cortical cardiomyocytes.

The list of genes to discriminate cortical from trabecular myocardium in the zebrafish is short: besides the herein reported *tbx5a* expression pattern and *nppa*[29], no other markers have been described to differentiate cortical from trabecular myocardium in the zebrafish. To interrogate whether *tbx5a*-derived trabecular cardiomyocytes not only relocate into the cortical region and switch off *tbx5a* but also adopt the expression of cortical marker genes, we performed RNA sequencing (RNA-Seq) comparing *tbx5a*-positive and -negative cardiomyocytes isolated from adult ventricles (Fig. 7a, b) . *tbx5a* was the top gene differentially expressed between the two populations (Fig. 7c and Supplementary Data 1), further supporting that the transgenic *tbx5a* reporter lines reproduce the endogenous pattern. A number of genes related to actin cytoskeleton, extracellular matrix, and caveolae were expressed specifically in *tbx5a*− cardiomyocytes (Fig. 7c, d): two of these markers, *xirp2a*[30] and *lama5*, were validated by immunofluorescence for their protein product (Figs. 8 and 9). In uninjured hearts, Xirp2a was detected in the cortical myocardium (Fig. 8a–f, $n = 3/3$). Strikingly, the *tbx5a*-derived GFP+ cells that contributed to the cortical layer and switched off *tbx5a*:mCherry also expressed Xirp2a (Fig. 8g–r, $n = 6/6$). Laminin was detected in the cortical layer (Fig. 9a–f, $n = 3/3$) and, upon injury, cortically

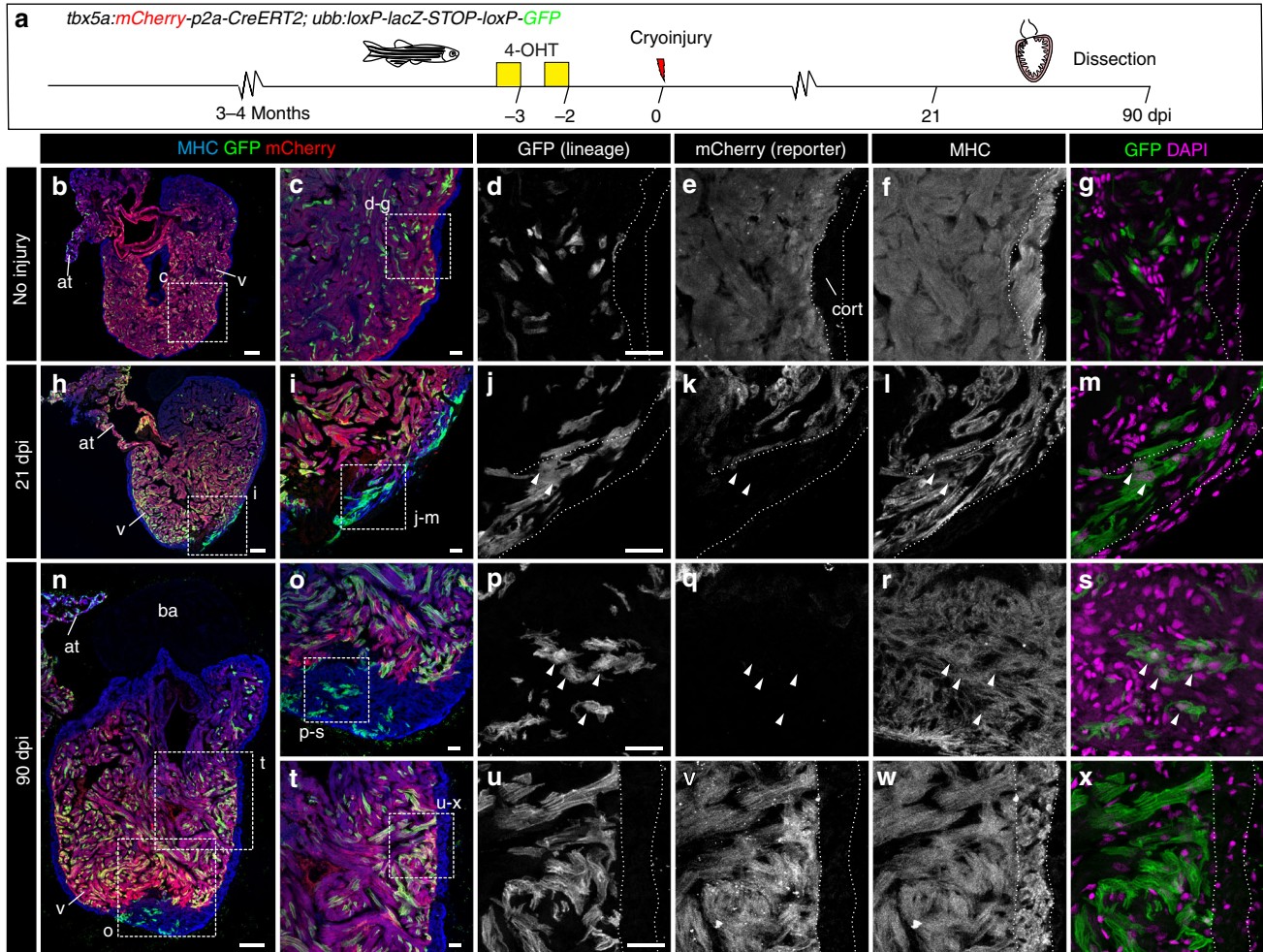

**Fig. 6** Contribution of *tbx5a*-derived cells during regeneration of the adult zebrafish heart. **a** *tbx5a:Cherry-p2A-CreER^{T2}* was crossed into *ubb:loxP-lacZ-STOP-loxP-GFP*. 4-Hydroxytamoxifen (4-OHT) was added 2 and 3 days before cryoinjury, to induce recombination of *loxP* sites. Hearts were fixed at 21 and 90 days postinjury (dpi) and sectioned for immunofluorescent detection of GFP^+ *tbx5a*-derived cells and mCherry^+ *tbx5a*-expressing cells. Nuclei were counterstained with DAPI. **b** In the uninjured heart (*n* = 7/7), mCherry expression was homogeneous in the trabecular myocardium and absent in the cortical layer. GFP^+ cells were found in the trabecular layer. **c** Zoomed view of boxed area in **b**. **d**–**f** Single channels of boxed area shown in **c**. **g** GFP and DAPI channels only. **h**, **n** Section of hearts at 21 dpi **h** and 90 dpi **n**. Upon cryoinjury to the ventricular apex, *tbx5a*-derived cardiomyocytes were present also in the cortical layer, particularly at the site of injury (**h**–**m**, *n* = 6/7; **o**–**s**, *n* = 5/6; **t**–**x**, *n* = 6/6), whereas *tbx5a*^+ cardiomyocytes in general were restricted to the trabecular myocardium (*n* = 5/6) **t**–**x**. Nuclear counterstaining revealed GFP^+ cell bodies in the cortical layer (arrowheads). at, atrium; cort, cortical layer; ba, bulbus arteriosus; v, ventricle. Scale bars, **b**, **h,n** 100 μm and **c**, **d**, **i**, **j**, **o**, **p**, **t**, **u** 25 μm

located *tbx5a*-derived GFP^+ were in a Laminin^+ environment; Fig. 9g–r, *n* = 6/6. This result was further supported using a RNAScope protocol to detect *lama5* mRNA in situ (Supplementary Fig. 15; a-h, *n* = 4/4; i-p, *n* = 11/11). A third gene used to determine whether *tbx5a*-derived cardiomyocytes express cortical markers when found in the cortical myocardium was *hey2*, a well-established marker of the compact layer in the embryonic mouse heart[31] (Supplementary Fig. 16). *hey2* mRNA in situ detection by RNAScope revealed expression in cardiomyocytes from the cortical myocardium of the adult zebrafish (Supplementary Fig. 16 a-h; *n* = 3/4). *tbx5a*-derived trabecular cardiomyocytes that were within the cortical region at 90 dpi co-localized with *hey2* staining (Supplementary Fig. 16 i-m; *n* = 5/6). Of note, *tbx5a*-derived cells within the trabecular myocardium did not reveal similar *hey2* expression levels (Supplementary Fig. 16 n-p; *n* = 5/6). On the contrary, the expression of the trabecular marker *nppa* was not retained in *tbx5a*-derived cells within the cortical myocardium (Supplementary Fig. 17, *n* = 7/7), altogether indicating that trabecular *tbx5a*^+ cardiomyocytes undergo a true phenotypic

switch to become cortical layer cardiomyocytes. Although the cortical myocardium had previously been reported to regenerate from cardiomyocytes within the same layer[5], our results reveal that trabecular cardiomyocytes can also contribute to cortical layer repair in response to injury.

## Discussion

In the developing hearts of birds and mammals, a border between *Tbx5*-positive and -negative cells determines the region of ventricular septation[32]. Our results here establish that the embryonic ventricle in zebrafish is concomitantly also split into *tbx5a*^+ and *tbx5a*^− domains. A similar boundary between *drl*^+ and *drl*^− cells has been described to separate the embryonic ventricle into domains to control distinct physiological properties[11], suggesting that similar to mammals, adult teleost cardiac ventricles maintain FHF and SHF lineage territories (Fig. 10a). Although the zebrafish ventricle is not septated, the presence of a FHF–SHF-like

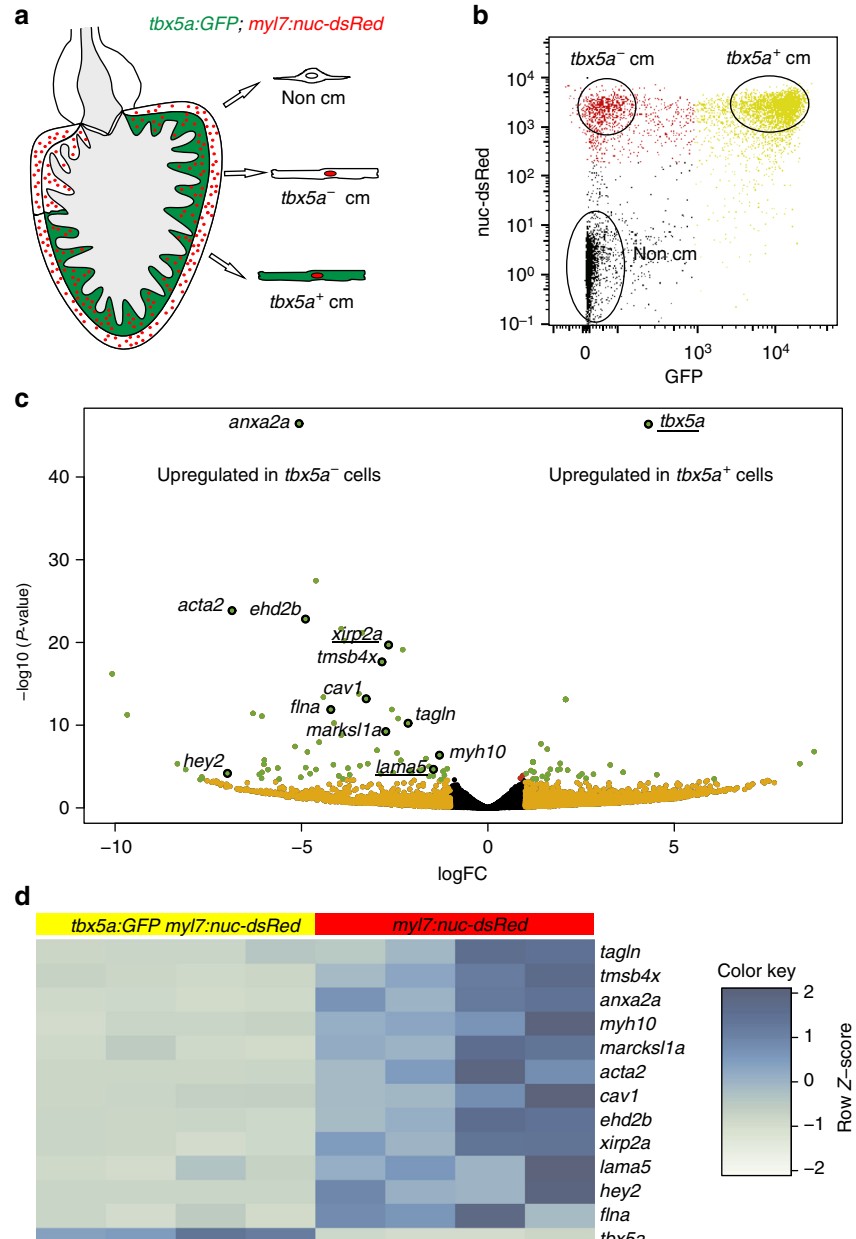

**Fig. 7** *tbx5a*[+] and *tbx5a*[−] cardiomyocytes from adult ventricles exhibit distinct expression profiles. **a**, **b** GFP[+]/nuc-dsRed[+] and GFP[−]/nuc-dsRed[+] cardiomyocytes were fluorescence-activated cell (FAC) sorted from adult *tbx5a:GFP;myl7:nuc-dsRed* ventricles (*n* = 5 pooled heart per replicate; four replicates in total). **c** Volcano plot representing RNA-seq results comparing both populations. Black, false discovery rate (FDR) > 0.05, log fold change (LFC) < 1; orange, FDR > 0.05, LFC > 1; red, FDR < 0.05, LFC < 1; green, FDR < 0.05, LFC > 1. **d** Heatmap of genes differentially expressed in *tbx5a*[+] and *tbx5a*[−] cardiomyocytes from adult hearts. Dark blue, higher expression; light blue, lower expression. cm, cardiomyocytes

boundary might represent a state that has been further progressed in evolution to allow morphogenetic events leading to septation.

Our results indicate that cardiomyocyte plasticity allows the ventricle to re-form exclusively from SHF-derived cardiomyocytes (Fig. 10b). Of note, in mammals, *Tbx5* expression labels the derivatives of the FHF in the ventricle and also the boundary with the SHF[19,20]; thus, we cannot exclude that the contribution of the SHF is slightly bigger than described in our study. Nonetheless, our results show that SHF cardiomyocytes can compensate the loss of the whole FHF-derived ventricle.

*tbx5a* lineage tracing allowed (i) to confirm that the cortical layer forms during development from the trabecular myocardium[5] and (ii) to identify that this process involves not only a change in location of cardiomyocytes from the trabecular to the cortical regions but also a change in gene expression (Fig. 10a).

In contrast to development, cardiomyocyte plasticity seems to be reduced in the regenerating adult heart:[19] conversion of atrial cardiomyocytes to ventricular cardiomyocytes has not been conclusively reported[7] and Gupta et al.[5] described that the newly regenerated cortical layer derives from pre-exising cortical layer cardiomyocytes. Indeed, the general rule during regeneration is that every cell contributes to the same cell type it was before injury, as illustrated by the cell types comprising the axolotl limb[33]. In contrast, our results show that a breach of this behavior is possible, with trabecular cardiomyocytes re-specialising into cortical layer cardiomyocytes during regeneration of the adult zebrafish heart (Fig. 10c). This re-specialisation reuses a

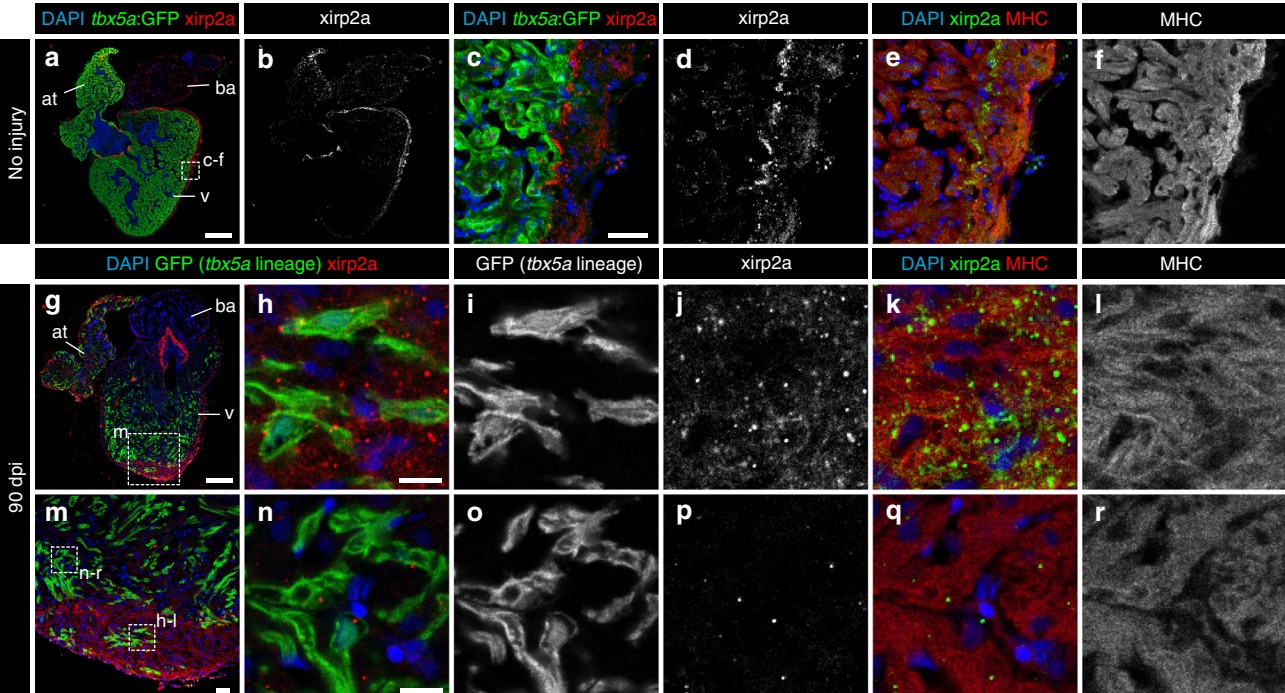

**Fig. 8** Trabecular *tbx5a*-derived cardiomyocytes within the cortical myocardium express the cortical marker Xirp2a. Immunofluorescence with anti-Xirp2a and GFP and anti-Myosin Heavy Chain (MHC) on ventricle sections. **a–f** Uninjured adult *tbx5a:GFP* ventricle. Xirp2a expression was observed in the cortical layer but not in the trabecular layer showing a complementary pattern with *tbx5a:GFP* as predicted by the RNA-Seq (n = 3/3). **g–r** Double transgenic *tbx5a: mCherry-p2a-CreER^T2^;ubb:loxP-lacZ-loxP-GFP* were treated with 4-Hydroxytamoxifen (4-OHT) from 84 to 72 and 60 to 48 hours before cryoinjury. Hearts were fixed at 90 days postinjury (dpi). GFP+ cells marking the *tbx5a* lineage within the cortical layer were positive for Xirp2a, whereas GFP+ cells within the trabecular layer did not express this marker (n = 6/6). at, atrium; ba, bulbus arteriosus; v, ventricle. Scale bars, **a**, **g** 100 μm, **c**, **m** 25 μm, and **h**, **n** 10 μm

developmental process in the setting of regeneration to rebuild the injured heart. It is interesting to note that zebrafish hearts are highly trabeculated in comparison with adult mammals. In murine cardiogenesis, the trabecular myocardium is proliferating at a much lower rate than the compact myocardium[34]. Thus, the switch from a trabecular phenotype to cortical phenotype might additionally allow the cardiomyocytes to re-enter a more active rate of cycling, allowing them to drive the regeneration after cryoinjury in the adult context. We propose that cardiomyocyte plasticity is thus likely a key feature necessary to react efficiently with a regenerative response to injury.

## Methods

**Animal handling and generation of transgenic lines.** Ethical approval was obtained from the Community of Madrid "Dirección General de Medio Ambiente" in Spain and the "Amt für Landwirtschaft und Natur" from the Canton of Bern, Switzerland. Animals were housed and experiments performed in accordance with Spanish and Swiss bioethical regulations for the use of laboratory animals. Fish were maintained at a water temperature of 28 °C.

The construct to generate Tg(*tbx5a:tdTomato*) transgenic zebrafish lines was made by recombining the tdTomato cassette (Supplementary Data 2) into the BAC CH73-99A14. The construct for Tg(*tbx5a:mCherry-p2A-CreER^T2^*)cn4 was generated by recombining the *iTol2Amp* cassette[35] (Supplementary Table 1, primers 1,2) and *mCherry-p2A-CreER^T2^* (Supplementary Data 3 and Supplementary Table 1, primers 3,4) into the same BAC. The construct to generate Tg(*tbx5a:CreER^T2^*)cn3 was made by recombining *iTol2Amp-γ-crystallin:RFP* (Supplementary Data 4 and Supplementary Table 2, primers 2,5) and CreER^T2^ (Supplementary Table 1; primers 4,6) cassettes into the same BAC. Tg(*vmhcl:loxP-myctagBFP-STOP-loxP-NTR-mCherry*)cn5 was generated using a construct obtained from recombining *iTol2Amp*[35] (Supplementary Table 1, primers 1,2) and *loxP-myctagBFP-STOP-loxP-mCherry-NTR* (Supplementary Data 5 and Supplementary Table 2, primers 7,8) cassettes into the BAC CH73-204E19. Plasmid templates for recombineering were cloned using Gibson Assembly (NEB). Recombineering was performed combining the *pRedET* (GeneBridges, Germany) system and EL250 bacteria[36]. Tg(-3.5ubb:loxP-lacZ-loxP-eGFP)cn2 [26] was outcrossed for six generations to isolate the best insertion. BAC DNA was injected at 25 ng μl⁻¹ into one-cell stage zebrafish

embryos along with 1 nl of 50 ng μl⁻¹ synthetic *Tol2* mRNA in Danieau buffer. Transient *ltbp3:TagRFP-2A-Cre* embryos in the stable *tbx5a:GFP* background were generated by injecting a Tol2 plasmid containing the vector *ltbp3:TagRFP-p2A-Cre*[12] into one-cell stage Tg(*tbx5a:GFP*) embryos. Around 150 embryos survived the microinjection and were screened for mCherry expression. The transgenic line *drl:mCherry* ^zh705^ (in full Tg(-6.3drl:mCherry)^zh705^) based on transgene vector pCM330 was generated using Multisite Gateway assembly of pCM293 (pENTR5' backbone containing 6.35 kb of the zebrafish *drl* locus (ZDB-GENE-991213-3) amplified with primers 5'-GTCAGCACCAGATGCCTGTGC-3' (forward) and 5'-CCAAGTGTGAATTGGGATCG-3' (reverse) as described[37], Tol2kit[38] #386 (pME-mCherry), #302 (p3E_SV40polyA), and #394 (pDestTol2A2) (in total pDestTol2pA2_drl:mCherry, referred to as *drl:mCherry*); plasmid DNA was injected at 25 ng μl⁻¹ into one-cell stage zebrafish embryos that were then raised and screened for germline transmission of the transgenic reporter with subsequent outcrossing to isolate a single transgene insertion[39]. The *drl:mCherry* transgenics used in the study are at least seventh-generation transgenics. The line Tg(-26.5Hsa. WT1-gata2:EGFP)cn1 [40] contains a reporter construct flanked by *flippase recognition target* sites, in which cardiac actin drives the expression of RFP. This cassette was removed by injection of *flippase* mRNA into the one-cell stage. This line was called Tg(-26.5Hsa.WT1-gata2:EGFP)cn12 and was replacing Tg (-26.5Hsa.WT1-gata2:EGFP)cn1.

In adults, 10 μM 4-OHT (Sigma, H7904) was administered at the indicated times. Treatments were performed overnight. Before administration, the 10 mM stock (dissolved in ethanol) was heated 10 min at 65 °C[39]. For genetic labeling in *tbx5a:mCherry-p2a-CreER^T2^;3.5ubb:loxP-lacZ-loxP-GFP* embryos, 4-OHT was administered at 10 μM from 24 to 48 hpf or at 5 μM from 48 to 84 hpf. Cryoinjury was performed as previously described using a copper filament cooled in liquid nitrogen and placed on the ventricular surface of the heart until thawing[41]. The pericardial cavity had been previously opened to expose the heart, in anesthetized animals. For genetic ablation experiments and their controls, 4-OHT was administered at 5 μM from 24 to 48 hpf and then Mtz (Sigma, M3761) was added at 10 mM from 96 to 168 hpf. Other lines used were TgBAC(*tbx5a:GFP*)in2Tg;[24] Tg(*myl7:nDsRed*)f2[42] Tg(*myl7:membranemCherry*)[43], Tg(3.5ubb:loxP-mCherry-loxP-eGFP)[44], and Et(-26.5Hsa.WT1-gata2:EGFP)cn1 (*epi:GFP*)[40].

For all the experiments involving Cre recombination or cell ablation, hemizygous fish were used. For experiments using only the Tg(*tbx5a:GFP*) line, homozygous and hemizygous fish were used interchangeably.

BrdU was added to E3 water at 5 mg ml⁻¹ with 0.5% dimethyl sulfoxide from 4 to 7 dpf.

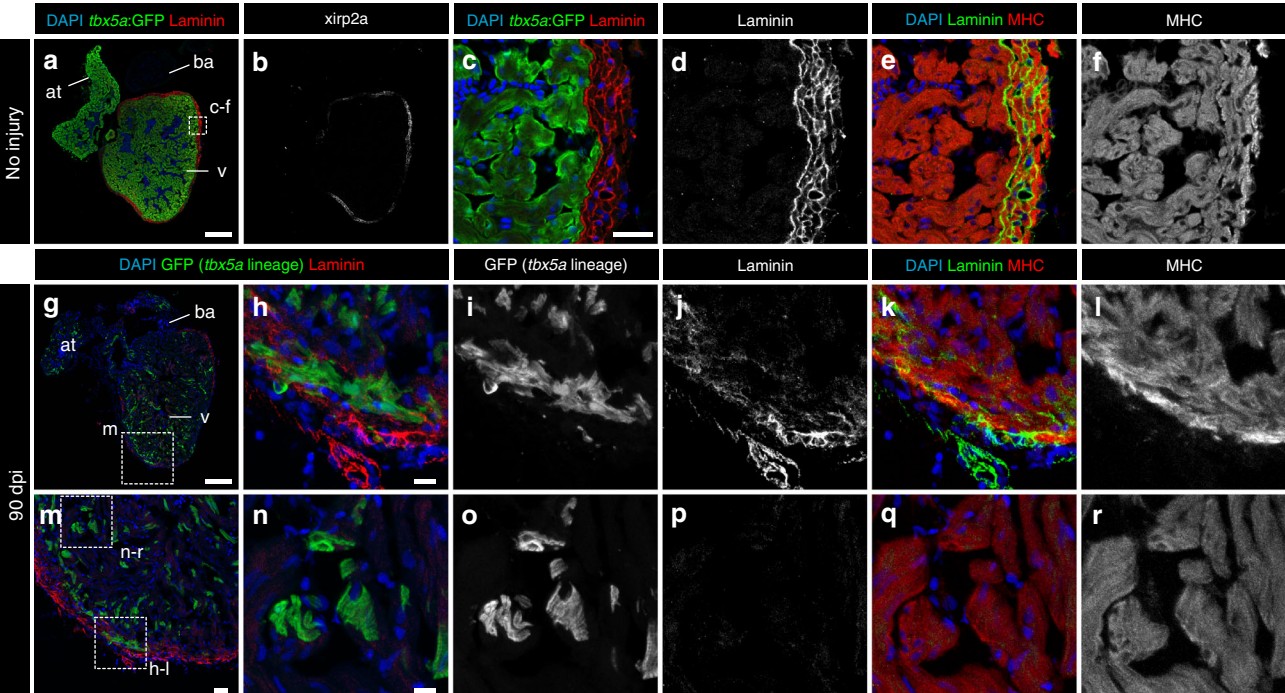

**Fig. 9** Trabecular *tbx5a*-derived cardiomyocytes within the cortical myocardium are surrounded by Laminin. Immunofluorescence with anti-Laminin, anti-GFP, and anti-Myosin Heavy Chain (MHC) on ventricle sections. **a–f** Uninjured adult *tbx5a:GFP* ventricle. Laminin expression was observed in the cortical layer but not in the trabecular layer showing a complementary pattern with *tbx5a:GFP* (*n* = 3/3). **g–r** Double transgenic *tbx5a:mCherry-p2a-CreER^T2^;ubb:loxP-lacZ-loxP-GFP* adult fish were treated with 4-Hydroxytamoxifen (4-OHT) from 84 to 72 and 60 to 48 hours before cryoinjury. Hearts were fixed at 90 days postinjury (dpi). GFP+ cells marking the *tbx5a* lineage within the cortical layer were surrounded by Laminin staining (*n* = 6/6). at, atrium; ba, bulbus arteriosus; v, ventricle. Scale bars, **a**, **g** 100 μm, **c**, **m** 25 μm, and **h**, **n** 10 μm

**Histology**. Samples for Fig. 2a Fig. 3a, and Supplementary Figures 1c–g, 3, 4, 13 and 17 were fixed in 4% pParaformaldehyde (PFA) in phosphate-buffered saline (PBS) overnight at 4 °C. Samples were then washed in 0.1% Tween20 (Merck) in PBS, dehydrated through an ethanol series, and embedded in paraffin wax. Samples were sectioned at 7 μm with a microtome (Leica), sections mounted on Superfrost slides (Fisher Scientific), and dried overnight at 37 °C.

The rest of sections were obtained by fixing in 4% PFA washing in PBS + 0.1% Tween20 (PBT), incubated in 15% saccharose overnight 4 °C. Then, samples were embedded in 30% gelatin 15% saccharose and snap frozen at − 80 °C in isopentane. Tissue was cut at 8 μm on a cryostat (Leica).

**In situ mRNA hybridization**. In situ mRNA hybridisation on paraffin sections and on whole-mount larvae were performed as described[45,46] using *tbx5a* (complementary DNA kindly provided by C. Neumann), *GFP* (cDNA kindly provided by J.L. Gómez-Skarmeta), and *nppa*[47] riboprobes.

Paraffin sections were deparaffinized, postfixed 20 min with PFA 4%, washed with PBS, treated with proteinase K 10 μm ml⁻¹ for 10 min at 37 °C, washed with PBS, postfixed with PFA 4% for 5 min, washed with PBS, treated with HCl 0.07 N for 15 min, washed with PBS, treated with 0.25% acetic anhydride in triethanolamine 0.1 M pH 8 for 10 min, washed with PBS, washed with RNase-free water, and then hybridised with the probe in pre-hybridisation buffer over night at 65 °C. The following day, sections were washed twice with post-hybridisation buffer 1 (50% formamide, 5 × SSC, 1% SDS) for 30 min at 65 °C and twice more with post-hybridisation buffer 2 (50% formamide, 2 × saline-sodium citrate SSC, 1% SDS). Then, sections were washed with maleic acid buffer with 0.1% Tween 20 (MABT) buffer at room temperature and incubated at least 2 h in blocking solution at room temperature. They were next incubated overnight with anti-dig-alkaline phosphatase (AP) antibody (1 : 2,000) in blocking solution. The third day, they were washed with MABT for several hours, followed by washes in AP buffer. Sections were developed in BM-Purple (Roche) until signal was detected.

For whole-mount in situ hybridisations, embryos were fixed in 4% PFA overnight, dehydrated in methanol series, and stored at − 20 °C until its use. On day 1, embryos were rehydrated in PBT, digested with 10 μg ml⁻¹ for 15 min, rinsed in PBT, and fixed in 4% PFA for 10 min. After washing again in PBT, they were pre-hybridised at 67 °C for at least 1 h. The antisense riboprobe was added at 0.5 μg ml⁻¹. After overnight hybridisation, two washes with 50% formamide/2 × SSCT, one wash with 2 × SSCT, and an additional two washes with 0.2 × SSCT were performed, all at 67 °C. Then, embryos were transferred to room temperature and washed in MABT. Antibody incubation, washing, and color development condition times were as for in situ on sections. RNAScope (Advanced

Cell Diagnostics, Hayward, CA) was performed following the manufacturer's instructions for formalin-fixed paraffin-embedded samples with standard tissue pretreatment and 2.5 HD RED detection kit.

**Immunofluorescence and TUNEL in whole-mount embryos**. Embryos were fixed in 4 % PFA overnight, washed in 0.1% Tween20 in PBS, and permeabilised with 0.5% Triton X-100 (Sigma) in PBS for 20 min. Several washing steps were followed by 2 h of blocking with 5% goat serum, 5% bovine serum albumin (BSA), and 20 mM MgCl₂ in PBS, followed by incubation with antibodies overnight. Apoptosis was detected by TUNEL staining using the in situ cell death detection kit from Roche (Mannheim, Germany), incubating fish embryos in enzyme solution for 1 h[48].

**Immunofluorescence on sections**. Paraffin sections were deparaffinized, rehydrated, and washed in distilled water. Epitopes were retrieved by heating in 10 mM citrate buffer (pH 6.0) for 15 min in a microwave at full power. Gelatine sections were incubated instead 30 min in 0.1% Tween20 in PBS at 37 °C to dissolve the gelatin. Nonspecific binding sites were saturated by incubation for at least 1 h in blocking solution (5% BSA, 5% goat serum, and 20 mM MgCl₂ in PBS). Endogenous biotin was blocked with the avidin–biotin blocking kit (Vector, Burlingame, CA, USA).

**Antibodies**. Primary antibodies used were anti-Myosin Heavy Chain (MF20, DSHB, 1 : 20 and F59, DSHB, 1 : 20), anti-GFP (AVES, GFP-1010, 1 : 500; 632592, Clontech, Mountain View, CA, USA, 1 : 100), anti-RFP (ab34771, AbCam, 1 : 200), anti-Xirp2a[30], a kind gift from C. Otten and S. Seyfried (1 : 500), anti-Laminin (L9393, Sigma, 1 : 200), anti-BrdU (BD Biosciences, B44, 1 : 100), anti-mKate[49] to detect tagBFP (1 : 500). Biotin- or Alexa (488, 568, 633)-conjugated secondary antibodies, and streptavidin-Cy3 (Jackson Immuno Research Laboratories) were used at 1 : 300. Nuclei were stained with 4′,6-Diamidino-2-phenylindole (DAPI) and slides were mounted in Fluorsave (Calbiochem).

**Imaging and Image analysis**. Embryos were imaged with a Zeiss 780 confocal microscope using a × 20 objective with a dipping lens. Z-stacks were taken every 1 μm. Three-dimensional images were reconstructed with ImageJ software. A Leica TCS SP-5 or a Nikon A1R confocal microscope was used for imaging of histological sections. The percentage of tagBFP cells was quantified using ImageJ considering the area of tagBFP and comparing it with the area of myosin heavy chain (MHC) staining.

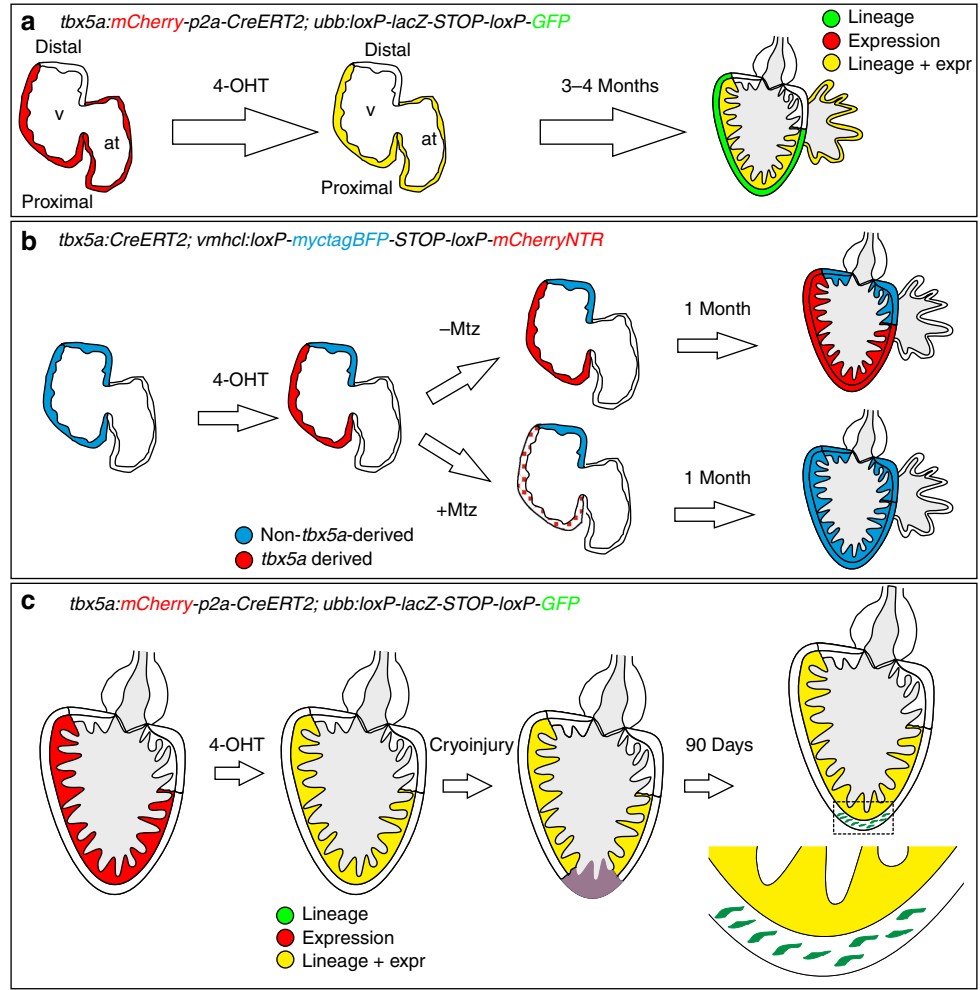

**Fig. 10** Summary of the contribution of *tbx5a*-positive and -negative myocardium during heart development and regeneration. **a** Identification of *tbx5a*-expressing and *tbx5a*-derived cells in the zebrafish heart. Red, *tbx5a*⁺ cells; green, *tbx5a*-derived cells not expressing *tbx5a*; yellow, *tbx5a*-expressing cells derived from embryonic *tbx5a*⁺ cells. **b** Replacement of the embryonic first heart field  myocardium with second heart field  progenitors. Blue, ventricular cardiomyocytes; red, *tbx5a*-derived ventricular cardiomyocytes; dotted red, ablated *tbx5a*-derived ventricular cardiomyocytes. **c** Contribution of *tbx5a*-derived cells during heart regeneration in the zebrafish. Red, *tbx5a*⁺ cells; green, *tbx5a*-derived cells not expressing *tbx5a*; yellow, *tbx5a*-expressing cells derived from trabecular adult *tbx5a*⁺ cells. 4-OHT, 4-Hydroxytamoxifen; at, atrium;  Mtz, Metronidazole; v,ventricle

The percentage of *tbx5a*-derived cells was quantified applying a median filter of radius 1 in ImageJ. The GFP⁺ and MHC⁺ areas were measured. The same set of images was used to quantify the percentage of GFP⁺ cardiomyocytes in trabeculae and in the regenerated compact layer, applying the same threshold for both regions.

**In vivo imaging and image processing**. Double transgenic Tg(*tbx5a:tdTomato*);Et (*-26.5Hsa.WT1-gata2:EGFP*) lines were grown until 60 hpf in E3 culture medium with 0.0033% 1-Phenyl-2thiourea. Embryos were anesthetized with 0.08 mg ml⁻¹ Tricaine and embedded in 1 % low melting agarose (Bio-Rad Low Melt Agarose). Confocal imaging of the embryonic heart was performed with a Zeiss LSM880, using a ×20/0.8 air-objective lens. The green (GFP) and red (tdTomato) channels were acquired simultaneously with maximum speed in bidirectional mode. Z-stacks were acquired over 8 h with 10 min time interval. The range of detection for each channel was adapted to avoid any crosstalk between the channels. A median filter was applied to a representative frame to show co-expression of the epicardial marker and *tbx5a*.

**Ultrasound and assessment of cardiac function in larvae**. Measurements and analysis were performed as described[47]. A mixture of 60 mM of tricaine and 3 mM of isofluorane were added to the tank water in order to anesthetize the animals. A Petri dish was filled with this solution and the fish was placed ventral side up in a foam holder. Once the animal was anesthetised, the images were obtained with a VEVO 2100 system (VisualSonics) coupled to a 50 MHz ultrasound transducer. The probe was lowered until an ultrasound conductive film was formed with the water, so that the skin of the animal was not touched at any time. This way the animal position was rotated till we obtained a long axis view of the heart with a complete visualisation of the ventricular apex. In that moment we recorded cine two-dimensional sequences, with a frame rate of 200 images per second, which were analysed later, in a second step, to minimize anesthesia times.

Imaging of cardiac function in larvae was performed using a Leica AM TIRF MC microscope in epifluorescence mode, using an  Andor DU 885K-CS0-#VP (1,004 × 1,002) camera and HCX PL APO 20 × 0,7 dry (11,506,166) objective. The maximum (diastolic) and minimum (systolic) ventricular areas were measured in order to determine ventricular function using Image J.

**Heart dissociation followed by sorter and RNA-Seq library production**. Zebrafish hearts were dissected and the atrio-ventricular canal was carefully removed in order to obtain a pool of exclusively ventricular cells. The ventricles were dissociated according to previous protocols[50] with minor modifications. The enzyme concentration was doubled and time of digestion increased to 1 h and 40 min with gentle agitation while pipetting with a cut 1,000 µl tip every 20–30 min. Then, one volume of PBS + 10% fetal bovine serum (FBS) was added and the mixture was centrifuged for 8 min at 250 g and re-suspended in PBS + 1% FBS. Specifically, the following enzyme concentrations were used: liberase TH (Sigma, 200 mg l⁻¹), Elastase (Serva, 1 : 250), Pronase E (Serva, 1 : 100 dilution of the 26.3 mg ml⁻¹ stock), DNase (Qiagen, 1 : 20,000) and 2,3-butanedione monoxime (Sigma, 10 mM). All reagents were dissolved in Tyrode's low calcium (in mM): NaCl 140, KCl 5.4, CaCl₂ 0.01, MgCl₂ 1.0, glucose 5.5, and HEPES 5.0; pH was set to 7.4 with NaOH.

Cells were sorted using a SONY Synergy sy3200 sorter and RNA was extracted using the Arcturus Pico Pure RNA isolation kit (Thermofisher) following the manufacturer's instructions.

Total RNA (0.6 ng) was used to generate barcoded RNA-Seq libraries using the Ovation Single Cell RNA-Seq System (NuGEN) with two rounds of library amplification. The size of the libraries was calculated using the Agilent 2100 Bioanalyzer. Library concentration was determined using the Qubit fluorometer (ThermoFisher Scientific). Libraries were sequenced on a HiSeq2500 (Illumina) platform to generate 60-base single reads. FastQ files for each sample were obtained using CASAVA v1.8 software (Illumina). Four biological replicates consisting of five pooled hearts were used per sample.

**RNA-Seq analysis**. Sequencing adaptor contaminations were removed from reads using cutadapt 1.9.1 software[51] and the resulting reads were mapped and quantified on the transcriptome (Ensembl gene-build 10, release 82) using RSEM v1.2.25[52]. Only genes with at least one count per million in at least two samples were considered for statistical analysis. Data were then normalised and differential expression tested using the bioconductor package EdgeR[53]. We considered as differentially expressed those genes with a Benjamini–Hochberg adjusted $p$-value ≤ 0.05 and log fold change ≥ 1. A heatmap was made using gplot library and heatmap.2 function.

**Data availability**. RNA-Seq data have been deposited in the GEO database under accession code GSE87596. The authors declare that all data supporting the findings of this study are available within the article and its supplementary information files or from the corresponding author upon reasonable request. Raw data corresponding to the Figures are deposited at Mendeley: https://doi.org/10.17632/fkm2tvh2sv.1; https://doi.org/10.17632/4fp5f8t7j3.1 and https://doi.org/10.17632/zgdff55m44.1

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

## Acknowledgements

We are grateful to the Animal facility, Histology, Microscopy, Cellomics, Bioinformatics, and Genomics Units from CNIC, the Microscopy Imaging Center from the University of Bern; X. Langa, L. Flores, M. Villalba, and R. Costa for experimental assistance; S. Seyfried and C.-B. Chien for reagents; and M. Torres, S. Martin-Puig, J. González-Sainz-de-Aja, R.M. Benedito, and A. Jazwinska for discussion. Funding was through FPU12/03007 and BFU2014–56970–P (Plan Estatal de Investigación Científica y Técnica y de Innovación 2013–2016, Programa Estatal de I+D+i Orientada a los Retos de la Sociedad Retos Investigación: Proyectos I+D+i 2016, del Ministerio de Economía competitividad e Industria), and co-funding by Fondo Europeo de Desarrollo Regional (FEDER) (H.S. and N.M.); Swiss National Science Foundation grant 31003A_159721, ANR-SNF collaborative Project 320030E-164245, and the ERC starting grant 337703–zebra–Heart (N.M.). A. E. is enrolled into PhD specialization Cutting Edge Microscopy offered by the Graduate School for Cellular and Biomedical Sciences (GCB) and the Microscopy Imaging Center (MIC). The CNIC is supported by the Ministry of Economy, Industry and Competitiveness (MEIC) and the Pro CNIC Foundation, and is a Severo Ochoa Center of Excellence (MEIC award SEV-2015-0505). Further support was from the Canton of Zürich, project grant from the Swiss Heart Foundation (A.F. and C. Mosimann); the Swiss National Science Foundation (SNSF) professorship (PP00P3_139093) and a Marie Curie Career Integration Grant from the European Commission (CIG PCIG14-GA-2013-631984) (C. Mosimann).

## Author contributions

H.S.-I. designed and carried out experiments. M.G.-C., A.S.-M and A.E. carried out experiments. G.G.-M. carried out ultrasound experiments. A.F. and C. Mosimann generated the *drl:mCherry* line. C.M. generated the *tbx5a:GFP* and *tbx5a:tdTomato* lines. J.M.G.-R. contributed in the generation of transgenic lines. N.M. designed experiments and secured funding. All authors contributed to writing the manuscript.

## Additional information

**Competing interests:** The authors declare no competing financial interests.

