## [Peer Review File · Nature Communications]

Reviewers' Comments:

Reviewer #1:

Remarks to the Author:

This is an interesting paper from the Mercader lab that addresses issues of the origin and plasticity of cardiomyocytes during zebrafish heart regeneration.

Lineage tracing and inducible ablation experiments are often not so straightforward to interpret, and a number of points need to be addressed.

Major comments.

The authors combined the *tbx5a* reporter line with an established SHF marker (*Itbp3*) to provide evidence that *tbx5a* expression serves as a bona fide marker for first heart field identity during early cardiogenic stages. The data obtained in larvae at 72hpf look convincing as the expression of *Itbp3* and *tbx5a* do not show overlapping expression (Extended Data Figure 1).

Shortly thereafter during development, *tbx5a* expression loses its properties as a first heart field marker and starts to be highly enriched in the trabecular myocardium as shown at 5dpf (Figure 1j-l). Unfortunately, the authors do not show any intermediate stages, ie, between 72hpf (prior to trabeculation) and 5dpf. Given the stability of GFP, this switch might occur very early in development. The trabecular specific expression of *tbx5a* remains static until adulthood. As all trabecular cells are developmental derivatives of cells of the compact wall, this shows that *tbx5a* expression is dynamic at larval stages. This is also evidenced and mentioned by the authors when comparing *tbx5a* expression in adult hearts with *tbx5a* based lineage tracing, which differ from each other (Figure 2 l-n).

To test a potential level of plasticity of FHF and SHF during regeneration, the authors now use a combination of *tbx5a* driven 4-OHT inducible CRE and *vmhc1:lox-bfp-lox-cherry-NTR* to convert and ablate FHF derived myocardial cells. Based on the results, the authors conclude that the ablation of roughly 95% of the ventricular myocardium (FHF derived) is compensated by massive expansion of the minor 5% SHF population.

It appears highly unlikely that a zebrafish larva could survive the loss of 95% of its functional ventricular myocardium. The authors note in their rebuttal letter that zebrafish larvae are able to survive without circulation (*tnnt2a* mutant) for up to a week which is somewhat correct, but it is important to note that cardiac insufficiency leads to the development of severe pericardial edema and that larvae are usually not able to recover from a transiently stopped heart because of this edema.

A possible explanation for the observed result is that the authors ablated trabecular cells rather than first heart field myocardium which would most likely be lethal given the ratio of cells mentioned above. This could possibly happen if the 4-OHT remains active after washout as mentioned in previous studies (Akerberg et al 2014), *tbx5a* expression is higher in trabecular cells over initial FHF expression or if Cre induction by 4-OHT occurred late. The remaining (*tbx5a* negative) myocardium of the wall would sustain contractility in this case and rebuild trabeculation leading to the observed result. However, this would not provide evidence for plasticity between FHF and SHF as claimed in the manuscript.

Another possibility is that surviving embryos recombined poorly therefore observing in the adults mostly tBFP+ cells (Fig. 2t-y). Therefore, to rule out this possibility recombination and ablation should be documented by imaging the same fish at the relevant times during the course of the experiment, to show efficient recombination and ablation during early developmental stages. Importantly, the interpretation of these results still is not conclusive. The absence of mCherry+ cells only would indicate no atrial contribution. Again, contribution by the so-called SHF-derived cells is not directly tested.

Given how critical this point is for the overall conclusion of the manuscript, I would highly recommend to quantify the cell ablation by total numbers and by myocardial subtype (wall or trabecular cell). This should be feasible given the tools the authors have in hand. Also, a quantification of cardiac performance after ablation and recovery would be helpful to better

interpret the results.

In adults, *tbx5a* expression is trabecular specific. Lineage tracing of *tbx5a* positive cells through cardiac regeneration after cryoinjury results in the presence of formerly *tbx5a* positive cells within newly formed myocardium. The authors have now used antibody staining to test the hypothesis that those cells not only lost their trabecular fate but obtained a novel cell fate within the regenerate (Figure 3 and 4). It would be most helpful if the authors could show separate channels for red green and blue in the relevant closeups of figure 4 to allow the reader to conclude independently. The loss of trabecular fate is convincing but it remains unclear which cell fate these cells adopted.

Fig. 3n: despite the fact that a MHC marker is used, this heart does not seem very representative and, as other reviewers suggested, authors should use another heart or 130 dpi as end point since, at least in their setup, it seems to be when the heart recovers (ED Fig. 7v-z). Importantly, the authors point to *tbx5a*:GFP⁺ epicardial cells in adult hearts (ED Fig. 7d-f) and these cells can also be observed during larval stages (Fig. 2c-e). Epicardial contribution during regeneration should be tested to rule out that *tbx5a*:GFP⁺ cells observed in the cortical wall after regeneration do not correspond to epicardial derived cells. Moreover higher quality images and with higher magnification should be shown to facilitate the interpretation of the data. Mef2 staining will be much more convenient to confirm CM specificity of the *tbx5a*:GFP⁺ cells in the regenerate rather than MHC. Co-localization with epicardial markers should also be shown at larval stages.

other comments

Data supporting the existence of a *tbx5a*:GFP⁻ region is poorly supported. There are examples where, when the single channel is shown (e.g. ED Fig. 7c,h,r,w), no *tbx5a*:GFP⁻ domain is observed. These are not the only examples in the manuscript, for instance the same applies to data shown in ED Fig. 2a. Since this is one of the main claims, this subpopulation should be documented and characterized much more carefully, and not only relying on one transgenic line. In addition, the location of *tbx5a*:GFP⁻ CMs is confusing, especially in view of the new data provided in ED Fig. 4 and ED Fig. 5. In ED Fig. 4l-q *tbx5a*:GFP⁻ CMs are shown starting from the mid-transversal plane towards the apex, while in ED Fig. 5 *tbx5a*:GFP⁻ CMs are found close to the base of the bulbus arteriosus. To characterize this sub-population of CMs more specific tools are required rather than absence of *tbx5a*:GFP expression, which cannot be consistently observed throughout the data provided in the manuscript. SHF markers should be used in adults for the characterization of this subpopulation.

tbx5a:GFP⁺ CM sorting. These expression data should be validated using a different approach not relying on the *tbx5a*:GFP line (e.g. LCM). In addition antibodies used for validation do not show convincing cortical expression. Laminin seems to be in the cortical interstitium, and Xirp2a is hardly over background levels. Co-localization with Mef2 Ab, high magnification and individual channel images should be shown in all the cases. Also ISH for the best candidates could be performed to confirm the cortical specificity.

- ED Fig. 2 *tbx5a* ISH should be done in hearts from wild-type instead of *tbx5a*:GFP fish. In addition ISH for *gfp* in hearts from *tbx5a*:GFP fish would help to further support the specificity of the expression pattern displayed by the transgenic line.

- The number of *tbx5a*:GFP-derived cells in the cortical myocardium should be quantified and colocalized with Mef2 Ab.

Line 168, data on mortality rate should be provided.

- Line 226, these data should be shown.

- Lines 261-263, this claim is not correct, please check Jensen et al. PlosOne 2012

- Lines 338, 339, the number of *tbx5a*:GFP-derived cells present in the cortical myocardium is generally quite reduced. How meaningful is this to regeneration? This evidence should be discussed.

Reviewer #3:

Remarks to the Author:

The authors have carried out the requested changes and included improved images and additional data, and provided logical responses for the proposed comments/suggestions. They have also changed the ambiguous/controversial article title and included a discussion on the subject. The suggested changes have improved the article, and consequently this work can now be recommended for publication in Nature Communications.

Reviewers' comments:

Reviewer #1 (Remarks to the Author):

This is an interesting paper from the Mercader lab that addresses issues of the origin and plasticity of cardiomyocytes during zebrafish heart regeneration. Lineage tracing and inducible ablation experiments are often not so straightforward to interpret, and a number of points need to be addressed.

Major comments.

*The authors combined the *tbx5a* reporter line with an established SHF marker (*ltp3*) to provide evidence that *tbx5a* expression serves as a bona fide marker for first heart field identity during early cardiogenic stages. The data obtained in larvae at 72hpf look convincing as the expression of *ltp3* and *tbx5a* do not show overlapping expression (Extended Data Figure 1).*

We are happy that the reviewer found this data convincing.

*Shortly thereafter during development, *tbx5a* expression loses its properties as a first heart field marker and starts to be highly enriched in the trabecular myocardium as shown at 5dpf (Figure 1j-l). Unfortunately, the authors do not show any intermediate stages, ie, between 72hpf (prior to trabeculation) and 5 dpf. Given the stability of GFP, this switch might occur very early in development. The trabecular specific expression of *tbx5a* remains static until adulthood. As all trabecular cells are developmental derivatives of cells of the compact wall, this shows that *tbx5a* expression is dynamic at larval stages. This is also evidenced and mentioned by the authors when comparing *tbx5a* expression in adult hearts with *tbx5a* based lineage tracing, which differ from each other (Figure2 l-n).*

In the early heart tube, the myocardium is an epithelial tube. Cells from this epithelium grow into the lumen and form the trabecules starting around 3 dpf (Liu et al., 2010). Several weeks later, in the ventricle, trabecular cells breach the outer epithelial-like myocardial layer, named primordial layer, and the peripheral cortical layer starts enveloping the ventricle.

As noted by the reviewer, the expression of *tbx5a* during heart development is dynamic. We have included an intermediate stage and would like to briefly explain the expression pattern. At 32 hpf *tbx5a*:GFP expression is observed in the whole myocardial epithelium (Fig. 1a-c). At 56 hpf, *tbx5a*:GFP is expressed in the myocardial epithelium with the exception of the cells at the arterial pole (Fig. 1d-f). We call this region the distal myocardium, while the *tbx5*:GFP positive part of the ventricle is named proximal ventricular myocardium.

At 3 and 4 dpf (newly included stage) *tbx5a*:GFP is expressed in proximal part of the ventricle and atria (Fig. 1g-l). Note that the expression of GFP is higher in the forming trabecular myocardium than in the primordial layer.

At 5 dpf the expression pattern is similar to 4 dpf (Fig. 1m-o).

The expression pattern of anti-GFP has been validated by mRNA *in situ* hybridization with a GFP riboprobe, as suggested by the reviewer. We do not see a mayor difference in the expression patterns (ED Fig. 1), thus stability of the GFP protein does not seem to present a problem for the interpretation of the results.

*To test a potential level of plasticity of FHF and SHF during regeneration, the authors now use a combination of *tbx5a* driven 4-OHT inducible CRE and *vmhcl:lox-bfp-lox-cherry-NTR* to convert and ablate FHF derived myocardial cells. Based on the results, the authors conclude that the ablation of roughly 95% of the ventricular myocardium (FHF derived) is compensated by massive expansion of the minor 5% SHF population.*

*It appears highly unlikely that a zebrafish larva could survive the loss of 95% of its functional ventricular myocardium. The authors note in their rebuttal letter that zebrafish larvae are able to survive without circulation (*tnnt2a* mutant) for up to a week which is somewhat correct, but it is important to note that cardiac insufficiency leads to the development of severe pericardial edema and that larvae are usually not able to recover from a transiently stopped heart because of this edema.*

When we treat *tbx5a:CreERT²; vmhcl:lox-bfp-lox-cherry-NTR* with 4-OHT, we observe that both, the primordial layer and the trabeculae, are efficiently recombined (Fig. 4c). We have now highlighted this in the revised figure. Thus, it is not only the trabeculae that are potentially sensitized to Mtz treatment. We have performed additional experiments to report genetic ablation of *tbx5a*-derived cardiomyocytes and their effect on heart regeneration. TUNEL assay on larvae treated with Mtz revealed staining of *tbx5a*-derived cardiomyocytes also within the primordial layer (Fig. 4j-n). Thus, primordial layer cardiomyocytes are also eliminated with this treatment. Quantification revealed a significant increase of TUNEL⁺ *tbx5a*-derived cardiomyocytes upon Mtz treatment (Fig. 4o). We observed that several cardiomyocytes rounded up and were extruded to the apical surface upon Mtz treatment. This mechanism of cell death has been shown to preserve epithelial tissue integrity (Gu and Rosenblatt, 2012). This might be an explanation why this cell ablation is not affecting survival of the animals.

To further investigate our conclusion, we performed the same cell ablation experiment but with *myl7:CreERT²; vmhcl:lox-bfp-lox-cherry-NTR* larvae, which trigger NTR expression in the whole myocardium. These experiments resulted in severe pericardial edema at the end of Mtz treatment (Fig. rev 1). This result confirms that the *tbx5a*-negative myocardial reservoir is required to compensate the loss of ventricular mass and that this compensation occurs rapidly after the induction of cell ablation or, most likely, concomitant with the ablation of *tbx5a*-positive cardiomyocytes.

Fig rev 1 | Comparison of the effect of cardiomyocyte ablation using a *tbx5a*- or pan-cardiomyocyte driver. **a**, Illustration of experimental set up. Embryos from the lines *tbx5a:CreERT2;vmhcl:loxP-tagBFP-STOP-loxP-mCherryNTR* (b) and *myl7:CreERT2^{pd10};vmhcl:loxP-tagBFP-STOP-loxP-mCherryNTR* (c) were treated with 4-Hydroxytamoxifen (4-OHT) from 24 to 48 hours postfertilization (hpf). Metronidazol (Mtz) was added from 4 to 7 days postfertilization (dpf). Larvae were imaged at 7 dpf. **b**, *tbx5a:CreERT2;vmhcl:loxP-tagBFP-STOP-loxP-mCherryNTR* revealed a normal phenotype (n=12/17). **c**, *myl7:CreERT2^{pd10};vmhcl:loxP-tagBFP-STOP-loxP-mCherryNTR* revealed pericardial effusion and edemas (n= 17/19). Scale bars, 100 μ m.

*A possible explanation for the observed result is that the authors ablated trabecular cells rather than first heart field myocardium which would most likely be lethal given the ratio of cells mentioned above. This could possibly happen if the 4-OHT remains active after washout as mentioned in previous studies (Akerberg et al 2014), *tbx5a* expression is higher in trabecular cells over initial FHF expression or if Cre induction by 4-OHT occurred late. The remaining (*tbx5a* negative) myocardium of the wall would sustain contractility in this case*

and rebuild trabeculation leading to the observed result. However, this would not provide evidence for plasticity between FHF and SHF as claimed in the manuscript.

Upon recombination, all ventricular cardiomyocytes in the proximal part are mCherry⁺ (see Fig. 3e; 4c; ED Fig 8j). We have highlighted recombined cells in the trabecular region and primordial region in Figure 3. If only trabecular cardiomyocytes would have recombined, we would see tagBFP⁺ staining in the primordial layer. Moreover, we detect TUNEL⁺ cardiomyocytes also in the outer myocardial layer. Finally, after cell ablation the myocardium changes from mostly mCherry⁺ to mostly tagBFP⁺. This observation goes against the hypothesis that Mtz administration leads to ablation of trabecular cells and their regeneration by primordial cells. In that case, the heart should be mCherry⁺ and not tagBFP⁺.

Another possibility is that surviving embryos recombined poorly therefore observing in the adults mostly tBFP⁺ cells (Fig. 2t-y). Therefore, to rule out this possibility recombination and ablation should be documented by imaging the same fish at the relevant times during the course of the experiment, to show efficient recombination and ablation during early developmental stages.

We have imaged larvae after 4-OHT treatment and before and after administration of Mtz (ED Fig. 9). Then, the animals were grown individually to adulthood and imaged again. Our longitudinal study confirms that upon embryonic 4OHT administration, recombination is efficient leading to mainly red hearts but that after Mtz treatment, the ventricle finally develops mainly from BFP⁺ cells.

Importantly, the interpretation of these results still is not conclusive. The absence of mCherry⁺ cells only would indicate no atrial contribution. Again, contribution by the so-called SHF-derived cells is not directly tested.

We agree that a SHF-CreERT2 line would be convenient as a separate proof but unfortunately, as mentioned in the first review round, such a line does not exist at this moment. The absence of mCherry⁺ cells reveals that there is a progenitor pool able to compensate for the lost *tbx5a*⁺ cardiomyocytes. Given that it is not the atrium (as these progenitors would be mCherry⁺), and that during regeneration myocardium has been shown to derive from pre-existent cardiomyocytes in the zebrafish (Jopling et al., 2010; Kikuchi et al., 2010), our results suggest that *tbx5a*⁻ cardiomyocytes contribute to heart regeneration. We observe that in Mtz-treated animals, tagBFP⁺ cells proliferate and expand, supporting that this distal ventricle population is expanding to compensate the loss of *tbx5a*-derived cardiomyocytes (Fig. 4p-u).

Moreover, when we ablate all ventricular cardiomyocytes at the same stage using *myl7:CreERT2:vmhcl:lox-bfp-lox-cherry-NTR*, the myocardium fails to recover (Fig. rev 1). Consequently, the small population of SHF cardiomyocytes in the ventricle is necessary for regeneration.

Given how critical this point is for the overall conclusion of the manuscript, I would highly recommend to quantify the cell ablation by total numbers and by myocardial subtype (wall or trabecular cell). This should be feasible given the tools the authors have in hand. Also, a quantification of cardiac performance after ablation and recovery would be helpful to better interpret the results.

We have performed TUNEL assay (Fig. 4j-o) and find a similar amount of apoptotic cells as previously described (Zhang et al., 2013). We were unable to accurately discriminate between primordial and trabecular TUNEL⁺ cardiomyocytes as the cells adopt a rounded morphology and loose there original location. In any case, we observe TUNEL⁺ cells in the periphery of the heart tube and we find that mCherry⁺ cells disappear, a good indication of their ablation.

In adults, tbx5a expression is trabecular specific. Lineage tracing of tbx5a positive cells through cardiac regeneration after cryoinjury results in the presence of formerly tbx5a positive cells within newly formed myocardium. The authors have now used antibody staining to test the hypothesis that those cells not only lost their trabecular fate but obtained a novel cell fate within the regenerate (Figure 3 and 4). It would be most helpful if the authors could show separate channels for red green and blue in the relevant closeups of figure 4 to allow the reader to conclude independently. The loss of trabecular fate is convincing but it remains unclear which cell fate these cells adopted.

We have followed the recommendation by the reviewer and included panels with the separate channels (Fig. 7). Please see also the Figures ED Fig. 12-13 showing *lama5* and *hey2* as additional cortical markers and *nppa* as a trabecular marker being downregulated (ED Fig. 14).

Fig. 3n: despite the fact that a MHC marker is used, this heart does not seem very representative and, as other reviewers suggested, authors should use another heart or 130 dpi as end point since, at least in their setup, it seems to be when the heart recovers (ED Fig. 7v-z).

Please note that the trabecular contribution to the new myocardium during regeneration is also shown in the ED Fig. 11s-w. This heart corresponds to the time point shown in 3n, with the only difference that the 4-OHT was added earlier. In both of them complete regeneration and trabecular to cortical transition can be detected. Additionally, we provide as a response to referees more examples (Fig. rev2). If needed, we can replace the image with any of these.

Fig rev 2 | *tbx5a*-derived cells contribute to the regenerating cortical myocardium. **a**, Experimental set up. **b-c, f-i** and **j-m** are three examples of hearts different from those shown in the main manuscript in which GFP⁺ cardiomyocytes can be observed in the cortical myocardium. Arrowheads point to the *tbx5a*-derived cells that have switched off *tbx5a*. Scale bars, 25 μ m.

*Importantly, the authors point to *tbx5a*:GFP⁺ epicardial cells in adult hearts (ED Fig. 7d-f) and these cells can also be observed during larval stages (Fig. 2c-e). Epicardial contribution during regeneration should be tested to rule out that *tbx5a*:GFP⁺ cells observed in the cortical wall after regeneration do not correspond to epicardial derived cells.*

Several studies have ruled out the epicardium as a source of cardiomyocytes during regeneration (Kikuchi 2011, Gonzalez-Rosa 2012, Jopling 2010 Kikuchi 2010).

Moreover higher quality images and with higher magnification should be shown to facilitate the interpretation of the data.

Please view the original TIFF files, which are of very high quality and provide good cellular resolution.

*Mef2 staining will be much more convenient to confirm CM specificity of the *tbx5a:GFP*+ cells in the regenerate rather than MHC.*

We decided to use MHC instead of Mef2c as co-localizing two cytoplasmic signals (a sarcomeric protein and the cytoplasmic GFP) allows to clearly assess if a cell is expressing both markers. Furthermore, in our hands, anti-Mef2 also labels non-cardiomyocytes (Fig. rev3).

Fig rev 3 | Mef2 immunofluorescent staining in the regenerating zebrafish heart. Zoomed view of the injury area of a cryoinjured heart 7 days postinjury. Shown is a sagittal section. Myosin Heavy Chain (MHC) marks cardiomyocytes, DAPI is used as nuclear counterstain. Anti-Mef2 (Santa Cruz Biotechnology, C-21) staining marks nuclei of MHC positive and negative cells. Scale bars, 100 μ m.

Co-localization with epicardial markers should also be shown at larval stages.

We have crossed a new line not reported in the previous manuscript, *tbx5a:td-Tomato*, with *epi:GFP*, a line we previously showed to specifically mark epicardial cells (Peralta et al., 2013). We found that *tbx5a* is expressed in a subset of epicardial cells (ED Fig. 5).

other comments

*Data supporting the existence of a *tbx5a:GFP*- region is poorly supported. There are examples where, when the single channel is shown (e.g. ED Fig. 7c,h,r,w), no *tbx5a:GFP*- domain is observed. These are not the only examples in the manuscript, for instance the same applies to data shown in ED Fig. 2a. Since this is one of the main claims, this subpopulation should be documented and characterized much more carefully, and not only relying on one transgenic line. In addition, the location of *tbx5a:GFP*- CMs is confusing, especially in view of the new data provided in ED Fig. 4 and ED Fig. 5. In ED Fig. 4l-q *tbx5a:GFP*- CMs are shown starting from the mid-transversal plane towards the apex, while in ED Fig. 5 *tbx5a:GFP*- CMs are found close to the base of the bulbus arteriosus. To characterize this*

*sub-population of CMs more specific tools are required rather than absence of *tbx5a:GFP* expression, which cannot be consistently observed throughout the data provided in the manuscript.*

We apologize that the domain is still unclear to the reviewer. We would like to point out that we were able to characterize this region with two transgenic lines, *tbx5a:GFP* and *tbx5a:mcherryP2ACreERT²*. The *tbx5a⁺* area is small and is not visible in all the sections. Depending on the orientation, it might be visible or not. We have now included arrowheads in all relevant Figures to illustrate the domain (see Fig. 2; ED Fig.4; ED Fig. 6, ED Fig.7). Note that the *tbx5:GFP⁺* trabecular cells in the basal region can be detected also in ED Fig. 10b,l, ED Fig. 11b, h, ED Fig 12a and ED Fig. 13a.

SHF markers should be used in adults for the characterization of this subpopulation.

Unfortunately, there are to date no well-established transgenic SHF markers in the zebrafish. We used *ltgfb3:tagRFP* as a marker (Zhou et al., 2011); we decided to use reproducible injection-based assays as the original stable transgenic BAC line has silenced. A lineage tracing of SHF-derived cells is not possible, as there is no inducible Cre line (CreERT2) available.

**tbx5a:GFP⁺* CM sorting. These expression data should be validated using a different approach not relying on the *tbx5a:GFP* line (e.g. LCM).*

In our view, RNASeq after sorting is a better technique than LCM, as with LCM it is not possible to separate cardiomyocytes from other cell types, leading to incorrect conclusions. For example, in Li et al, the authors report *tbx18*, a well-known epicardial marker, as the best marker for the compact layer (Li et al., 2016). This is likely an artefact that we do not have in our experiment, as we do not detect expression of endocardial/epicardial markers in the sorted cells.

*In addition antibodies used for validation do not show convincing cortical expression. Laminin seems to be in the cortical interstitium, and *Xirp2a* is hardly over background levels. Co-localization with *Mef2* Ab, high magnification and individual channel images should be shown in all the cases. Also ISH for the best candidates could be performed to confirm the cortical specificity.*

We have included single channels for *Xirp2* and Laminin immunofluorescence stainings (Fig. 7). We have not performed co-localization with *Mef2*, as in our hands this antibody also labels non-cardiomyocytes (Fig. rev 3). However, we have expanded the number of markers. We included RNAScope to detect *lama5* mRNA, and show that *lama5* expression can be found in cortical cardiomyocytes including *tbx5a*-derived cells (ED Fig. 12). Furthermore, we present data on *hey2*, revealing the same findings (ED Fig. 13). Moreover, we performed *in situ* hybridization with the trabecular marker *nppa* and found that in the trabecular, but not the cortical layer, *nppa* expression is found in *tbx5a*-derived cells (ED Fig. 14).

- ED Fig. 2 *tbx5a* ISH should be done in hearts from wild-type instead of *tbx5a:GFP* fish. In addition ISH for *gfp* in hearts from *tbx5a:GFP* fish would help to further support the specificity of the expression pattern displayed by the transgenic line.

We have included these control experiments (see ED Fig. 1 and ED Fig. 3)

- The number of *tbx5a:GFP*-derived cells in the cortical myocardium should be quantified and colocalized with *Mef2* Ab.

We have quantified the number of *tbx5a* derived cells in the cortical myocardium: We found that 7 ± 4 % of *tbx5a*⁺ cells are GFP positive. We prefer to include co-localization with MHC as shown in Fig. 5 as our GFP signal is cytoplasmic and therefore a nuclear signal such as *Mef2* is less convenient to show co-localization in sections. Furthermore, in our hand *Mef2* staining also marks non-cardiomyocytes (Fig. rev 3).

Line 168, data on mortality rate should be provided.

We have included the data. See lines 155-156 and ED Fig. 9j.

- Line 226, these data should be shown.

We have included this data on ED Fig. 11a-f.

- Lines 261-263, this claim is not correct, please check Jensen et al. PlosOne 2012

We thank the reviewer for drawing our attention to this article, which is now cited on line 212. We also included this marker for the analysis of the phenotype of *tbx5a*-derived cells (ED Fig. 14).

- Lines 338, 339, the number of *tbx5a:GFP*-derived cells present in the cortical myocardium is generally quite reduced. How meaningful is this to regeneration? This evidence should be discussed.

We thank the reviewer for suggesting this quantification, as it is important data for the paper. Note that although the percentage of GFP⁺ cells in the cortical myocardium is low (7 ± 4 % of the regenerated cortical cardiomyocytes), this does not mean that the contribution is so low, as it has to be related to the recombination efficiency (10 ± 6 %). Thus, trabecules might contribute to approximately 67 ± 39 % of the regenerated compact layer. We have included this discussion in lines 196-200.

Reviewer #3 (Remarks to the Author):

The authors have carried out the requested changes and included improved images and additional data, and provided logical responses for the proposed comments/suggestions. They have also changed the ambiguous/controversial article title and included a discussion on the subject. The suggested changes have improved the article, and consequently this work can now be recommended for publication in Nature Communications.

We thank the reviewer for having contributed to substantially increase the quality of our article.

** See Nature Research's author and referees' website at www.nature.com/authors for information about policies, services and author benefits

This email has been sent through the Springer Nature Tracking System NY-610A-NPG&MTS

Confidentiality Statement:

This e-mail is confidential and subject to copyright. Any unauthorised use or disclosure of its contents is prohibited. If you have received this email in error please notify our Manuscript Tracking System Helpdesk team at <http://platformsupport.nature.com>.

References

- Gu, Y., and Rosenblatt, J. (2012). New emerging roles for epithelial cell extrusion. *Curr Opin Cell Biol* 24, 865-870.
- Jopling, C., Sleep, E., Raya, M., Marti, M., Raya, A., and Belmonte, J.C. (2010). Zebrafish heart regeneration occurs by cardiomyocyte dedifferentiation and proliferation. *Nature* 464, 606-609.
- Kikuchi, K., Holdway, J.E., Werdich, A.A., Anderson, R.M., Fang, Y., Egnaczyk, G.F., Evans, T., Macrae, C.A., Stainier, D.Y., and Poss, K.D. (2010). Primary contribution to zebrafish heart regeneration by gata4(+) cardiomyocytes. *Nature* 464, 601-605.
- Li, G., Xu, A., Sim, S., Priest, J.R., Tian, X., Khan, T., Quertermous, T., Zhou, B., Tsao, P.S., Quake, S.R., *et al.* (2016). Transcriptomic Profiling Maps Anatomically Patterned Subpopulations among Single Embryonic Cardiac Cells. *Dev Cell* 39, 491-507.
- Liu, J., Bressan, M., Hassel, D., Huisken, J., Staudt, D., Kikuchi, K., Poss, K.D., Mikawa, T., and Stainier, D.Y. (2010). A dual role for ErbB2 signaling in cardiac trabeculation. *Development* 137, 3867-3875.
- Peralta, M., Steed, E., Harlepp, S., Gonzalez-Rosa, J.M., Monduc, F., Ariza-Cosano, A., Cortes, A., Rayon, T., Gomez-Skarmeta, J.L., Zapata, A., *et al.* (2013). Heartbeat-driven pericardiac fluid forces contribute to epicardium morphogenesis. *Curr Biol* 23, 1726-1735.
- Zhang, R., Han, P., Yang, H., Ouyang, K., Lee, D., Lin, Y.F., Ocorr, K., Kang, G., Chen, J., Stainier, D.Y., *et al.* (2013). In vivo cardiac reprogramming contributes to zebrafish heart regeneration. *Nature* 498, 497-501.

Zhou, Y., Cashman, T.J., Nevis, K.R., Obregon, P., Carney, S.A., Liu, Y., Gu, A., Mosimann, C., Sondalle, S., Peterson, R.E., *et al.* (2011). Latent TGF-beta binding protein 3 identifies a second heart field in zebrafish. *Nature* 474, 645-648.

Reviewers' Comments:

Reviewer #1:

Remarks to the Author:

The proposed key finding of the manuscript is a suggested plasticity between different myocardial subtypes (cardiomyocytes) during cardiac regeneration.

A principal source of misinterpretation comes from the dynamic expression of the transgenic tools utilized in this study. FHF cell ablation is performed under the control of the regulatory elements of *tbx5a* which are dynamic during development and do not remain FHF specific but rapidly become trabecular specific. Therefore, accurate and detailed data acquisition and presentation of the experimental outcome is essential to allow conclusive data interpretation.

Unfortunately, key control experiments have not been carried out thoroughly as requested:

Reviewers comment: Given how critical this point is for the overall conclusion of the manuscript, I would highly recommend to quantify the cell ablation by total numbers and by myocardial subtype (wall or trabecular cell). This should be feasible given the tools the authors have in hand. Also, a quantification of cardiac performance after ablation and recovery would be helpful to better interpret the results.

In response, the authors state they are unable to accurately quantify the ablation across cardiomyocyte subtypes, and seemingly did not assess cardiac performance after ablation. The performed TUNEL assay indicates apoptosis in only few cardiomyocytes, potentially very few cells of the myocardial wall. Additionally, the authors show in Fig. 4k that a substantial (majority?) number of mCherry-NTR+ cardiomyocytes is still present 48 hours after the initiation of the cell ablation procedure. Assessment of ventricular contractility after ablation of the entire ventricular *tbx5a*:GFP+ lineage and a detailed analysis and presentation of the performed cell ablation would be good indicators for successful ablation but has not been performed.

Overall, the experiment is not sufficiently analyzed to allow the claims drawn by the authors and leaves much room for interpretation (i.e. only trabecular cells are efficiently ablated or the majority of cardiomyocytes are only harmed but not killed which reduces their proliferative potential and allows SHF cells to 'overgrow' this lineage during development).

Important analysis:

- Recombination and ablation should, as mentioned before, be documented in detail by imaging the same fish before and directly after ablation as well as over the course of the entire experiment. The new data provided (ED Fig 9) lack images showing ablated CM after 4-OHT and how the tBFP+ CM population progressively expands over the course of the experiment. Without these data, poor recombination cannot be ruled out as the cause of the observed tBFP contribution after ablation (Fig. 4).

- Fig7 and ED Fig. 13. Data shown for validation of cortical specific expression is still not convincing. In the case of Xirp2 ab, it is hard to believe that the outer most Xirp2- tissue layer is the epicardium. As for laminin, as mentioned before, its expression seems to be restricted to the cortical interstitium. As previously requested, co-localization with a CM specific ab should be carried out and single channels shown. The authors claim that in their hands Mef2 ab also labels non CMs in the injured area but that seems not to be the case in the uninjured myocardium, therefore making this antibody suitable for the proposed experiments. Alternatively they could use MF20 staining. Similarly, *hey2* expression pattern does not seem to be cortical specific (ED Fig. 13b).

Referee #1

The proposed key finding of the manuscript is a suggested plasticity between different myocardial subtypes (cardiomyocytes) during cardiac regeneration. A principal source of misinterpretation comes from the dynamic expression of the transgenic tools utilized in this study. FHF cell ablation is performed under the control of the regulatory elements of *tbx5a* which are dynamic during development and do not remain FHF specific but rapidly become trabecular specific. Therefore, accurate and detailed data acquisition and presentation of the experimental outcome is essential to allow conclusive data interpretation.

We agree that a detailed description of the expression pattern of *tbx5a* allows a correct interpretation of the results. The expression pattern during development has been extensively characterized including the analysis of the expression pattern at additional stages as well as mRNA expression patterns as requested by the reviewer (Fig 1, ED Fig 1).

In the line *tbx5a:GFP*, cardiomyocytes of the early heart tube are GFP⁺ and from 32 onwards, GFP⁻ cardiomyocytes appear at the cranial pole of the heart. We call this region of *tbx5:GFP* negative myocardium the distal region. Within the proximal *tbx5a:GFP* positive region, GFP expression is visible in both the primordial layer and the newly forming trabecular layer. It is true that in the trabecular layer GFP immunostaining is stronger than in the primordial layer from 4 dpf onwards, but GFP mRNA expression can still be detected at 5 dpf, both at the level of GFP protein and mRNA (Fig. 5m and ED Fig 1f,g). Thus, our transgenic line based on the *tbx5a* regulatory region allows to distinguish cardiomyocytes from the primary heart tube from those added to cranial pole from 2 dpf onwards (corresponding to *tbx5a*⁻ SHF-derived cells).

Fig 1g,j,m shows expression of *tbx5*:GFP in the primordial and trabecular myocardium of the ventricle at 3,4 and 5 dpf. The distal GFP negative region is highlighted with an arrowhead.

ED Fig. 1f, g shows expression of GFP mRNA, which can be detected also in the primordial and trabecular myocardium, and being excluded from a distal domain (arrowhead).

Unfortunately, key control experiments have not been carried out thoroughly as requested:

Reviewers comment: Given how critical this point is for the overall conclusion of the manuscript, I would highly recommend to quantify the cell ablation by total numbers and by myocardial subtype (wall or trabecular cell). This should be feasible given the tools the authors have in hand. Also, a quantification of cardiac performance after ablation and recovery would be helpful to better interpret the results.

In response, the authors state they are unable to accurately quantify the ablation across cardiomyocyte subtypes,

Note that at the time of recombination, trabeculation has not started. The reviewer is right, in that the revised version only included a qualitative assessment of the type of ablated cardiomyocytes based on the recombination efficiency, which was indistinguishable between primordial and trabecular layer (Fig. 4c).

Figure 4c (from manuscript). *tbx5a*-derived cells are mCherry⁺ and the rest of ventricular cardiomyocytes are blue. Upper part corresponds to regions of the distal ventricle, lower part to the proximal ventricle. mCherry⁺ cardiomyocytes are found in the primordial layer (prim) and trabecular layer (trab).

We now include the data regarding the number of trabecular and primordial TUNEL⁺ cardiomyocytes after cell ablation; 70±20 % of the TUNEL⁺ cardiomyocytes from the ventricle are found within the primordial layer and 30±20 % in the trabecular layer (Fig. 4o). Thus, cardiomyocytes from both layers are ablated with our genetic approach.

TUNEL⁺ cells/heart

New Fig 4o. Quantification of TUNEL⁺ trabecular and primordial cardiomyocytes.

tbx5a⁺ ventricular cardiomyocytes were genetically ablated in *tbx5a:CreERT2;vmhcl:loxP-tagBFP-loxP-mCherry-NTR* double transgenic zebrafish. Recombination was induced by administration of 4-OHT at 1 and 2 dpf. Animals were divided into two groups. Cell ablation was induced in one group by administration of Metronidazol (Mtz) from 4 dpf onwards. Animals were fixed at 6 dpf, followed by TUNEL staining and immunofluorescence with anti-mCherry. TUNEL⁺, mCherry⁺ double-positive cells were counted in the whole ventricle. -Mtz (n= 8 fish embryos), +Mtz (n=8 fish embryos). Data are means ± SD; P < 0.001 by two-tailed t-test.

and seemingly did not assess cardiac performance after ablation.

We did not include these data as we understood that a functional assessment in the adult was requested (Fig 4i). Please find the requested data showing cardiac performance after ablation in larvae (ED Fig.12 and ED Video 6). Immediately after ablation of *tbx5a*-derived ventricular cardiomyocytes, the hearts did not pump as efficiently as those from the control group. In the videos, also note that the cardiomyocytes covering the ventricular surface are rounded, indicating that they are either dying or severely damaged.

ED Fig. 12a-e. Cardiac performance after ablation of *tbx5a*-derived cells. *tbx5a*⁺ ventricular cardiomyocytes were genetically ablated in *tbx5a:CreERT2;vmhcl:loxP-tagBFP-loxP-mCherry-NTR* double transgenic zebrafish. Recombination was induced by administration of 4-OHT at 1 and 2 dpf. Animals were divided into two groups. Cell ablation was induced in one group by administration of Metronidazol (Mtz) from 4 to 7 dpf. Videos were acquired at 7 dpf, immediately after the last Mtz treatment. **a-d** are still images from the videos from two Mtz (from a total of 4) and two non-treated animals (from a total of 3). Shown are lateral views of the heart, the head is to the left. The ventricle is outlines in yellow. Note the irregular shape and overall smaller area in Mtz-treated hearts. **e**, The maximum (diastolic) and minimum (systolic) ventricular area was measured in order to determine ventricular function. P= 0.0348 by a two-tailed t-test.

The performed TUNEL assay indicates apoptosis in only few cardiomyocytes, potentially very few cells of the myocardial wall.

We were able to perform TUNEL staining and show a significant increase in cell death upon cell ablation compared with controls (Fig. 4k-o). We agree that the numbers are low but significant. Please take into account that apoptosis is a very fast process. With a TUNEL assay, only cardiomyocytes dying at that precise moment are detected. In the literature we could find only qualitative data but no quantification of the number of TUNEL⁺ cells expected upon cardiomyocyte cell ablation in embryonic zebrafish hearts^{1,2}. For example in Curado et al, Nat Protoc Fig 3b,b' two TUNEL⁺ cardiomyocytes are highlighted in the ventricle and one in the atria, in a larvae in which NTR is expressed in the whole myocardium.

The quantification of TUNEL⁺ cells shown in new Fig 4o reveal that approximately 2/3 of the TUNEL⁺ cardiomyocytes are located in the primordial layer.

Additionally, the authors show in Fig. 4k that a substantial (majority?) number of mCherry-NTR+ cardiomyocytes is still present 48 hours after the initiation of the cell ablation procedure.

The reviewer is right that at 48 hours after initiation of Mtz treatment (6 dpf) we can still detect mCherry⁺ cardiomyocytes in the ventricle. However, please note that their morphology indicates damage. We chose this stage in order to still be able to still detect TUNEL⁺ cells. If we had waited

longer, we would have been unable to see mCherry⁺ cells. At 7 dpf, 72 hours after initiation of Mtz treatment, there are nearly no mCherry⁺ cells visible (new ED Fig. 10), so TUNEL staining was assessed at an earlier time point.

ED Figure 10. Assessment of efficient ablation of *tbx5a*-derived cardiomyocytes. *tbx5a*⁺ ventricular cardiomyocytes were genetically ablated in *tbx5a:CreERT2;vmhcl:loxP-tagBFP-loxP-mCherry-NTR* double transgenic zebrafish. Recombination was induced by administration of 4-OHT at 1 and 2 dpf. At 4 dpf, larvae were imaged under a binocular scope. Images show lateral views of the head and heart region. The lens is red (gamma-cristallin transgenesis reporter) and the ventricle is red (mCherry⁺ *tbx5a*-derived ventricular cardiomyocytes) and blue (*tbx5a*⁻ cardiomyocytes). Fish were individually treated with Mtz from 4 to 7 dpf. At 7 dpf, 3 days after initiation of Mtz treatment, each larva was imaged again. At this stage, tagBFP expression is visible and expanded but mCherry is no longer detected in most larvae. Scale bars, 100 μm.

Assessment of ventricular contractility after ablation of the entire ventricular *tbx5a:GFP*⁺ lineage and a detailed analysis and presentation of the performed cell ablation would be good indicators for successful ablation but has not been performed.

The following data has been included:

1. Quantitative assessment of trabecular versus primordial TUNEL⁺ cells (Fig. 4o)
2. Cardiac performance after ablation of *tbx5a*-derived cells (ED Fig. 12a-e and ED Video 6).

3. Longitudinal study using mCherry and tagBFP expression before and after Mtz treatment to show ablation of mCherry⁺ *tbx5a*-derived cardiomyocytes (ED Fig. 10).
4. Qualitative and quantitative assessment of the proportion of mCherry⁺ vs tagBFP⁺ cells in the adult ventricle (Fig. 4b-h).
5. Ultrasound assessment of cardiac function in adult zebrafish with and without cell ablation (Fig. 4i).
6. Qualitative assessment of recombination efficiency by cell type (Fig. 4c).
7. Qualitative and quantitative assessment of cell ablation by TUNEL (Fig. 4j-o).
8. Qualitative and quantitative assessment of proliferation of *tbx5a*⁺ and *tbx5a*⁻ derivatives after cell ablation (Fig. 4p-u).
9. Longitudinal study of 2 fish showing efficient recombination in the embryo and compensation of the *tbx5a*-derived myocardium in the adult by *tbx5a*⁻ cells (ED Fig. 11).
10. Assessment of fish survival in cell ablation experiments (ED Fig. 12f).

We hope the reviewer will find this a sufficiently detailed presentation and analysis of the cell ablation experiments performed, and will agree that it indicates successful ablation.

Overall, the experiment is not sufficiently analyzed to allow the claims drawn by the authors and leaves much room for interpretation (i.e. only trabecular cells are efficiently ablated or the majority of cardiomyocytes are only harmed but not killed which reduces their proliferative potential and allows SHF cells to ‘overgrow’ this lineage during development).

We present more evidence in Figure R1 for this important question to show that not only trabecular cardiomyocytes that are efficiently ablated.

There are some considerations that preclude the referee’s hypothesis that only trabecular cells are efficiently ablated.

We show that in 4-OHT-treated *tbx5a:CreERT2;vmhcl:loxP-tagBFP-loxP-mCherry-NTR* larvae, the trabecular and primordial layers within the proximal part of the ventricle are mCherry⁺ (Fig. 4c,ED 11b,f). If only the trabeculae were ablated, the primordial layer would still be mCherry⁺. This is not what we observe.

Figure R1. Scheme to illustrate that the cell ablation results do not support the hypothesis, that only trabeculae are ablated. The oval represents the ventricle. The inner part represents the trabeculae, the outer ring the primordial layer.

The reviewer points out the interesting possibility, that some cardiomyocytes might be harmed but not killed, reducing their proliferative potential and allowing SHF cells to 'overgrow' this lineage during development. We do not exclude this possibility and are happy to discuss it in the manuscript. Please note that this hypothesis does not invalidate our results and would still support our conclusion that SHF-derived cells can compensate the loss of healthy FHF-derived cardiomyocytes.

Important analysis:

- Recombination and ablation should, as mentioned before, be documented in detail by imaging the same fish before and directly after ablation as well as over the course of the entire experiment. The new data provided (ED. Fig 9) lack images showing ablated CM after 4-OHT and how the tBFP+ CM population progressively expands over the course of the experiment. Without these data, poor recombination cannot be ruled out as the cause of the observed tBFP contribution after ablation (Fig. 4).

We have included additional analysis of 7 fish to show efficient recombination and ablation in larvae (new ED Fig. 10). The longitudinal imaging of the cell ablation experiment presented in our report clearly shows a compensation of the mCherry⁺ population by tagBFP⁺ cells.

Please note that imaging a fish at different stages is technically very demanding. Every time a fish is imaged, we have to partially stop its heart, include it in agarose, and then remove it from the agarose and transfer it back to the fish water. This traumatic manipulation compromises fish survival and has not allowed us to examine the same fish at more than two stages. Zebrafish are not transparent throughout development, so that from 7 dpf onwards it is no longer possible to image the whole heart *in vivo* at cellular resolution. To our knowledge, imaging the whole heart at cellular resolution for this late period until adulthood has not been performed to date. A further difficulty is the working distance of objectives that has not allowed us to image the whole heart at this later time points. In our attempts to image tagBFP⁺ cardiomyocytes *in vivo* we observed an unexpected and very rapid fading of the fluorescence signal. We have therefore been unable to use *in vivo* imaging to show expansion of the tagBFP⁺ cardiomyocyte population in real-time.

The presence of tagBFP⁺ cells in the regenerated myocardium cannot be explained by poor recombination because of the following reasons:

- Only larvae showing efficient mCherry recombination were selected for the experiments. tagBFP⁺ cells are not present in the distal part of the adult ventricle in hearts from the -Mtz group (Fig. 4d), which would be a sign of poor recombination efficiency.

- In Fig 4r and ED Fig 10 we show that the SHF-derived cardiomyocytes have already started to expand to regenerate the lost *tbx5a*-derived myocardium shortly after starting the cell ablation. Proliferation of tagBFP cells in the distal ventricle was observed (Fig. 4r). A more scattered appearance of tagBFP⁺ cells would be expected if mCherry⁺ cells were compensated by proximally located unrecombined cells.

- Fig7 and ED Fig. 13. Data shown for validation of cortical specific expression is still not convincing. In the case of *Xirp2 ab*, it is hard to believe that the outer most *Xirp2*- tissue layer is the epicardium.

In response to the second revision of this reviewer, we included 2 additional cortical markers (*lama5*, *hey2*) as well as 1 trabecular marker (*nppa*).

We have now also included single and merged channels to show colocalization of Xirp2 with MF20. Note that Xirp2 colocalizes with MF20 in the cortical myocardium (Fig 7 e,f,k,l). The Figure from the manuscript has been updated with these additional panels (Fig. 7).

Figure 7. Trabecular *tbx5a*-derived cardiomyocytes within the cortical myocardium express Xirp2a. Immunofluorescence with anti-Xirp2a, GFP and Myosin Heavy Chain (MHC) on ventricle sections. **a-f**, Uninjured adult *tbx5a:GFP* ventricle. Xirp2a expression was observed in the cortical layer but not in the trabecular layer showing a complementary pattern to *tbx5a:GFP*, as predicted by the RNA-Seq. **g-r**, Double transgenic *tbx5a:mCherry-2a-CreER^{T2};ubb:loxP-lacZ-loxP-GFP* fish were treated with 4-OHT from 84h to 72h and 60h to 48-hours before cryoinjury. Hearts were fixed at 90 dpi. GFP⁺ cells marking the *tbx5a* lineage within the cortical layer were positive for Xirp2a. cm, cardiomyocytes. at, atrium; ba, bulbus arteriosus; dpi, days postinjury; v, ventricle.

As for laminin, as mentioned before, its expression seems to be restricted to the cortical interstitium. As previously requested, co-localization with a CM specific ab should be carried out and single channels shown. The authors claim that in their hands Mef2 ab also labels non CMs in the injured area but that seems not to be the case in the uninjured myocardium, therefore making this antibody suitable for the proposed experiments. Alternatively they could use MF20 staining. Laminin is an extracellular protein and as such is found in the interstitium, as noted by the reviewer. For this reason, we complemented these data with *lama5* RNAScope, to detect mRNA expression (now ED Fig. 15). We have included magnified views for Laminin and *lama5* stainings to show colocalization with MF20. The Figures from the manuscript have been updated (ED Fig. 15 and Fig. 8, respectively).

ED Figure 15. *tbx5a*-derived cardiomyocytes within the cortical layer are positive for *lama5*. GFP immunofluorescence and *lama5* RNAScope in situ detection on sagittal sections of *tbx5a:GFP* (a-h) or *tbx5a:mCherry-2a-CreER^{T2};ubb:loxP-lacZ-loxP-GFP* double transgenic fish, recombined before injury and fixed at 90 dpi (i-p). DAPI was used as a nuclear counterstain **a-d**, Uninjured heart. *tbx5a:GFP* (green) does not co-localize with *lama5* (red). *lama5* is expressed at higher and more homogeneous levels in the cortical layer. **e-h**, Uninjured heart. Anti-Myosin Heavy Chain (MHC) was used as a cardiomyocyte marker (green). In the cortical layer, regions of *lama5*/MHC co-localization were detected (white arrowheads). **i-p**, 90 dpi regenerated hearts. *tbx5a:mCherry-2a-CreER^{T2};ubb:loxP-lacZ-loxP-GFP* fish were treated with two 12 h pulses of 4-OHT 6 and 7 days before injury. GFP⁺ cells marking the *tbx5a* lineage within the cortical layer were positive for *lama5* (yellow arrowheads). GFP⁺ cells within the trabecular layer were negative for this marker (green arrowheads). at, atrium; ba, bulbus arteriosus; dpi, days postinjury; v, ventricle. Scale bars, 100 μ m (**a,e,i**), 25 μ m (**c,f,j,n**).

Figure 8. Trabecular *tbx5a*-derived cardiomyocytes within the cortical myocardium change marker gene expression. Immunofluorescence with anti-laminin, anti-GFP and anti-myosin heavy chain on ventricle sections. **a-f**, Uninjured adult *tbx5a:GFP* ventricle. Laminin expression was observed in the cortical layer but not in the trabecular layer showing a complementary pattern with *tbx5a:GFP*. **g-r**, Double transgenic *tbx5a:mCherry-2a-CreER^{T2};ubb:loxP-lacZ-loxP-GFP* fish were treated with 4-OHT from 84h to 72h and 60h to 48h before cryoinjury. Hearts were fixed at 90 dpi. cm, cardiomyocytes. at, atrium; ba, bulbus arteriosus; dpi, days postinjury; v, ventricle.

Similarly, *hey2* expression pattern does not seem to be cortical specific (ED Fig. 13b).

The reviewer is right: *hey2* is indeed not cortical specific but in the trabecular layer it is expressed in non-cardiomyocytes (please compare the expression of ED Fig. 16j with ED Fig 16n) as reported previously³. However this marker is well suited to distinguishing cortical from trabecular cardiomyocytes.

Extended Data Figure 16 | *tbx5a*-derived cardiomyocytes within the cortical layer are *hey2* positive. GFP immunofluorescence and *hey2* RNAScope in situ detection on sagittal sections of *tbx5a:GFP* (a-h) or *tbx5a:mCherry-2a-CreER^{T2};ubb:loxP-lacZ-loxP-GFP* double transgenic animals, recombined before injury and fixed at 90 dpi (i-p). DAPI is used as nuclear counterstain **a-d**, Uninjured heart. *tbx5a:GFP* (green) does not co-localize with *hey2* (red). *Hey2* is expressed at higher and more homogenous levels in the cortical layer. **e-h**, Uninjured heart. anti-Myosin Heavy Chain (MHC) is used as a cardiomyocyte marker (green). In the cortical layer, regions of *hey2*/MHC co-localization were detected (yellow arrowheads). **i-p**, 90 dpi regenerated hearts. *tbx5a:mCherry-2a-CreER^{T2};ubb:loxP-lacZ-loxP-GFP* were treated with two 12 hours pulses of 4-OHT at 6 and 7 days before the injury. GFP⁺ cells marking the *tbx5a* lineage within the cortical layer were positive for *hey2* (yellow arrowheads). GFP⁺ cells within the trabecular layer were negative for that marker (green arrowheads). at, atrium; ba, bulbus arteriosus; dpi, days postinjury; v, ventricle. Scale bars, 100 μm (a,e,i), 25 μm (c,f,j,n).

- 1 Curado, S., Stainier, D. Y. & Anderson, R. M. Nitroreductase-mediated cell/tissue ablation in zebrafish: a spatially and temporally controlled ablation method with applications in developmental and regeneration studies. *Nat Protoc* **3**, 948-954 (2008).
- 2 Zhang, R. *et al.* In vivo cardiac reprogramming contributes to zebrafish heart regeneration. *Nature* **498**, 497-501, doi:10.1038/nature12322 (2013).
- 3 Luxan, G., D'Amato, G., MacGrogan, D. & de la Pompa, J. L. Endocardial Notch Signaling in Cardiac Development and Disease. *Circ Res* **118**, e1-e18, doi:10.1161/CIRCRESAHA.115.305350 (2016).

Reviewers' Comments:

Reviewer #1:

Remarks to the Author:

The manuscript has been further improved and is now ready for publication in Nat. Comm.

REVIEWERS' COMMENTS:

Reviewer #1 (Remarks to the Author):

The manuscript has been further improved and is now ready for publication in Nat. Comm.

We thank the reviewer for this recommendation and for the input received which has improved the quality of the manuscript.